# PHYSICAL GRADIENTS FOR DEEP LEARNING

## ABSTRACT

Solving inverse problems, such as parameter estimation and optimal control, is a vital part of science. Many experiments repeatedly collect data and rely on machine learning algorithms to quickly infer solutions to the associated inverse problems. We find that state-of-the-art training techniques are not well-suited to many problems that involve physical processes since the magnitude and direction of the gradients can vary strongly. We propose a novel hybrid training approach that combines higher-order optimization methods with machine learning techniques. We take updates of a higher-order or domain-specific optimizer and embed them into the gradient-descent-based learning pipeline as a *physical gradient*, replacing the regular gradient of the physical process. This also allows us to introduce domain knowledge into training by incorporating priors about the solution space into the gradients. We demonstrate the capabilities of our method on a variety of canonical physical systems, showing that physical gradients yield significant improvements on a wide range of optimization and learning problems.

## 1 INTRODUCTION

Inverse problems that involve physical systems play a central role in computational science. This class of problems includes parameter estimation (Tarantola, 2005) and optimal control (Zhou et al., 1996). Solving inverse problems is integral in determining the age of the universe (Freedman et al., 2001), measuring the amount of dark matter (Padmanabhan & White, 2009), detecting gravitational waves (George & Huerta, 2018b), controlling plasma flows (Maingi et al., 2019), searching for neutrinoless double-beta decay (Agostini et al., 2013; Aalseth et al., 2018), reconstructing particle trajectories and energies (Belayneh et al., 2020), and testing general relativity (Dyson et al., 1920; Kraniotis & Whitehouse, 2003).

Decades of research in optimization have produced a wide range of iterative methods and algorithms for solving inverse problems (Press et al., 2007). Higher-order methods such as limited-memory BFGS (Liu & Nocedal, 1989) have been especially successful. These methods have two fundamental drawbacks. First, they require a good initial guess of the solution. When the initial guess is far away from the true solution in parameter space, iterative methods may diverge or converge to a local optimum instead. Second, their computational cost is strongly problem-dependent, as the gradient and possibly even Hessian matrix need to be evaluated for each iteration of the optimization. This may be negligible for simple tasks but when the optimization involves a physical simulation, each iteration can take seconds or longer. However, many experiments continuously or repeatedly collect data, yielding data sets with millions or billions of individual events, each posing an inverse problem. Solving all of these with iterative solvers poses a major challenge which is why, in such settings, data analysis has often turned to machine learning algorithms to quickly infer approximate solutions given the observations (Carleo et al., 2019; Delaquis et al., 2018; George & Huerta, 2018b; Agostini et al., 2013). The inferred solutions can then, for example, be used to filter data sets for interesting events or can serve as the starting point for a more accurate analysis (CMS Collaboration, 2020; George & Huerta, 2018a;b).

Interestingly, using machine learning models to solve the inverse problems also reduces the chances of divergence or convergence to undesired local optima. Unlike iterative solvers, learning models are jointly trained on large collections of data. The shared parameters allow updates from one example to avoid local minima from a different example. Fig. 1 shows a generic case illustrating this effect: The task is to localize a wave packet $A \cdot \sin(f \cdot x) \cdot exp(-\frac{1}{2}(x-x_0)^2/\sigma^2)$ and determine its amplitude from a noisy recorded time series (grey curve). Iterative solvers fail to reconstruct the wave packet

when the initial guess is not already close to the true parameters, even diverging in 0.12% of our cases. In contrast, a neural network trained on the same data using the same objective quickly learns to fit all wave packets. Details are given in appendix B.1. In this paper, we derive a hybrid optimization algorithm that integrates higher-order inverse solvers into the traditional deep learning pipeline without imposing these limitations and without discarding the progress made in first-order optimization. Our method can work with arbitrary second-order solvers when the solution space of the inverse problem allows for computation of the Hessian. Otherwise, it requires additional domain knowledge which is used to formulate the inverse solver in a more efficient manner. Our method computes the weight updates using both backpropagation as well as the embedded inverse-problem solver. Specifically, the first-order gradient of the physical process used in backpropagation is replaced by a *physical gradient* (PG) derived from the update of the inverse-problem solver. When used in conjunction with training deep learning models, physical gradients allow each update step to encode nonlinear physics, which yields substantially more accurate optimization directions. A key advantage of our approach is its compatibility with acceleration schemes (Duchi et al., 2011; Kingma & Ba, 2015) and stabilization techniques (Ioffe & Szegedy, 2015; Huang et al., 2016; Ba et al., 2016) developed for training deep learning models.

The presented hybrid training approach differs fundamentally from pure first-order training. While PGs take the place of regular gradients in the optimization algorithm, they are based on real physical states instead of the adjoint ones calculated in backpropagation. This feature makes it possible to integrate solution priors into the training without restricting the model architecture (Kim et al., 2019), adding soft constraints to the objective (Raissi et al., 2018), or relying exclusively on first-order updates (Rackauckas et al., 2020).

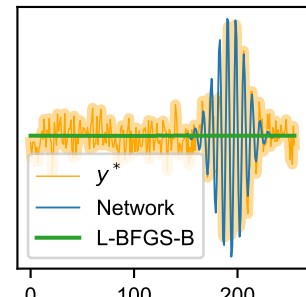

We test our method on a wide variety of inverse problems including the highly challenging Navier-Stokes equations. In all of our experiments, neural networks trained with PGs significantly outperform state-of-the-art training schemes, yielding accuracy gains of up to three orders of magnitude.

Figure 1: BFGS fails to fit a wave packet when the initial guess is too far from the true value. A neural network trained on the same objective learns to fit all examples.

## 2 METHOD

We consider unconstrained inverse problems that involve a differentiable physical process $\mathcal{P} : X \subset \mathbb{R}^{d_x} \to Y \subset \mathbb{R}^{d_y}$ which can be simulated. Here $X$ denotes the physical parameter space and $Y$ the space of possible observed outcomes. Given an observed or desired output $y^* \in Y$, the inverse problem consists of finding optimal parameters

$$x^* = \arg\min_x L(x) \quad \text{with} \quad L(x) = \frac{1}{2}\|\mathcal{P}(x) - y^*\|_2^2. \qquad (1)$$

We are interested in approximating this problem using a neural network, $x^* = \mathrm{NN}(y^* \,|\, \theta)$, parameterized by $\theta$. Let $\mathcal{Y}^* = \{y_i^* \,|\, i = 1, ..., N\}$ denote a set of $N$ inverse problems, all sharing the same physical process $\mathcal{P}$. Then training the network is equivalent to finding

$$\theta_* = \arg\min_\theta \sum_{i=1}^N \frac{1}{2}\|\mathcal{P}(\mathrm{NN}(y_i^* \,|\, \theta)) - y_i^*\|_2^2. \qquad (2)$$

This training scheme is unsupervised, i.e. no labels are required over the training data $\mathcal{Y}$.

### 2.1 ITERATIVE OPTIMIZATION

Inverse problems as in Eq. 1 are classically solved by starting with an initial guess $x_0$ and iteratively applying updates $\Delta x_k$. This yields an estimate of the inverse of the physical model $\mathcal{P}$, i.e., $x^* \approx \mathcal{P}_{n^*}^{-1}(y^* \,|\, x_0)$ where $n^*$ denotes the number of iterations required to reach a chosen convergence threshold. A key decision in such an iterative optimization scheme is how to derive the update $\Delta x_k$. Figure 2a visually compares gradient descent, Newton's method and PGs which we will introduce in section 2.2.

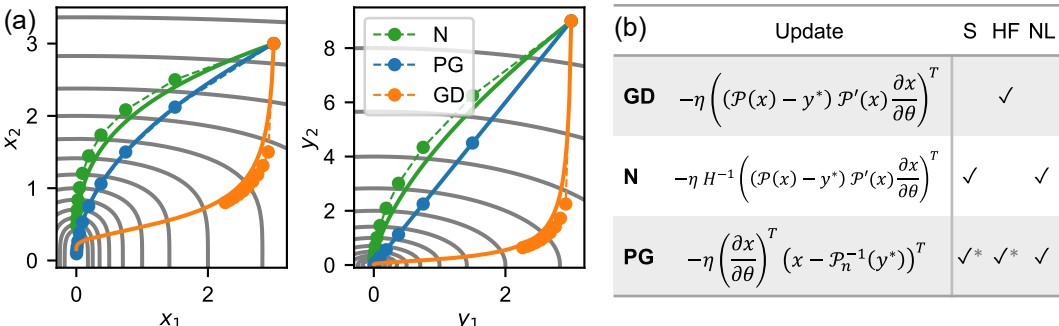

Figure 2: (a) Minimization of the function $L = \|y\|_2^2$ with $y = (x_1, x_2^2)$ in $x$ space. Contours of $L$ are shown in gray. Solid lines are the optimization trajectories of gradient descent (GD), Newton's method (N), and physical gradients (PG), with infinitesimal step sizes. Circles represent the first 10 iterations with constant step size. (b) Comparison of the same optimization methods by their respective update steps and properties: whether they can adapt to function sensitivity (S), are Hessian-free (HF) and take nonlinearities into account (NL). $H$ denotes the Hessian w.r.t. $\theta$. The properties of PG are explained in section 2.2.

**First-order methods**  The scale of typical machine learning datasets necessitates the use of mini-batches, which has led popular optimization methods in machine learning to rely only on first-order derivatives of $L$. These Hessian-free methods are popular in deep learning due to their low computational cost and stable convergence (Goodfellow et al., 2016). The simplest such method is gradient descent (GD) (Curry, 1944) and its stochastic version (SGD) where $\Delta x = -\eta \cdot \left(\frac{\partial L}{\partial x}\right)^T$. Note that $\frac{\partial L}{\partial x}$, and therefore of $\Delta x$, scales inversely with $x$. When $x$ carries physical units, neither $\Delta x$ nor $\Delta x^T$ lie in the same space as $x$. This scaling behavior is a problem when dealing with *sensitive* or *insensitive* functions, i.e. functions whose gradient magnitudes $\left|\frac{\partial L}{\partial x}\right|$ are far from 1. When small changes in $x$ cause large changes in $L$, this counter-intuitively leads to large updates $\Delta x$ when using GD, resulting in even larger changes to $L$. This is often referred to as exploding gradients, and the opposite effect, vanishing gradients, occurs with insensitive functions.  Machine learning models can be tuned to behave well given these updates (Ioffe & Szegedy, 2015; Loshchilov & Hutter, 2019) but in physics-based optimization, where functions may be extremely sensitive in certain parts of the parameter space or on subsets of training examples, GD often prescribes suboptimal optimization directions.

While there are a number of first-order optimizers that approximate higher-order information to improve convergence (Kingma & Ba, 2015; Hestenes et al., 1952; Duchi et al., 2011; Martens & Grosse, 2015), they cannot take nonlinearities into account, which results in suboptimal optimization directions when optimizing nonlinear functions.

**Second-order methods**  Newton's method (Atkinson, 2008) uses the inverse Hessian to determine the optimization update $\Delta x = -\eta \cdot \left(\frac{\partial^2 L}{\partial x^2}\right)^{-1} \left(\frac{\partial L}{\partial x}\right)^T$. Direct computation of the Hessian adds a significant computational cost to each iteration. Quasi-Newton methods (Broyden, 1970; Liu & Nocedal, 1989; Gill & Murray, 1978; Moré, 1978; Powell, 1970; Berndt et al., 1974; Conn et al., 1991; Avriel, 2003) alleviate this issue by approximating the inverse Hessian instead. Unlike GD, Newton-type methods prescribe small updates to sensitive functions and large updates to insensitive ones. The resulting update steps typically progress towards optima much faster than first-order methods (Ye et al., 2019). However, the difficulty of approximating the Hessian for batched data sets (Schraudolph et al., 2007) and the high cost of evaluating the Hessian directly have largely prevented second-order methods from being utilized in machine learning research (Goodfellow et al., 2016).

**Domain-specific inverse solvers**  Domain knowledge can often be used to formulate more efficient optimization updates or to directly approximate a solution. A classical example would be preconditioners that, when chosen correctly, allow linear solvers to converge much more quickly and reliably. In nonlinear settings, part of the governing equations may be inverted analytically or

priors about the solution space may be incorporated into the solver, for example. A time-reversed simulation may be able to more accurately recover prior states than a generic inverse solver. These methods are commonly used in optimization but their use has eluded machine learning applications so far due to the fixed gradient-descent pipeline. When learning with PGs, domain-specific optimizers can be integrated into training like any other optimizers, allowing for greatly-increased training convergence. Appendix appendix A.2 lists necessary conditions that need to be fulfilled.

## 2.2 PHYSICAL GRADIENTS FOR DEEP LEARNING

We now consider the problem of learning solutions to the inverse problems as in Eq. 2. As discussed above, GD-based optimizers fail to take the nonlinearity and potentially strongly varying sensitivity of the joint problem into account while higher-order optimizers are hard to apply directly.

Instead, we aim to leverage the advantages of higher-order optimizers in a deep learning setting by embedding arbitrary inverse-problem solvers as physical gradients (PGs) into the first-order optimization pipeline. The network updates $\Delta\theta$ can then be computed using any first-order optimizer such as SGD or Adam. Additionally, all state-of-the-art deep learning techniques, such as normalization Ioffe & Szegedy (2015); Ba et al. (2016) or dropout Srivastava et al. (2014) can be employed, which often improves training speed and generalization performance Keskar et al. (2017).

Our method can be applied whenever one of the following is available: (i) $\frac{\partial^2 \mathcal{P}}{\partial x^2}$ can be computed numerically, (ii) a higher-order update such as $\left(\frac{\partial^2 \mathcal{P}}{\partial x^2}\right)^{-1} \frac{\partial \mathcal{P}}{\partial x}$ can be derived analytically, or (iii) an inverse-physics solver can be derived using the available domain knowledge. If (i) is fulfilled, we can explicitly compute the Newton direction in in $x$ space. Thus, with any of (i), (ii) or (iii) we have access to better updates $\Delta x$ than $\Delta x_{\text{GD}} \propto \frac{\partial \mathcal{P}}{\partial x}$. These $\Delta x$ are the basis for our method which we describe in the following.

We derive our method starting from Eq. 2 which describes the combined problem of finding both solutions $x^*$ to the individual inverse problems $y^* \in \mathcal{Y}^*$ and network weights $\theta$ that approximate them. One way to solve this problem is to first consider all inverse problems separately and precompute corresponding solutions $\mathcal{X}^{\text{sv}} = \{x_i^{\text{sv}} : \mathcal{P}(x_i^{\text{sv}}) = y_i^*\}$ using an iterative optimizer. Then, $\mathcal{X}^{\text{sv}}$ could be used as labels for supervised training of a neural network:

$$\theta_* = \arg\min_{\theta} \sum_{i=1}^{N} \frac{1}{2} \|\text{NN}(y_i^* \mid \theta) - x_i^{\text{sv}}\|_2^2. \tag{3}$$

This precomputation makes it possible to use an efficient optimization method for $\mathcal{X}^{\text{sv}}$ while training the neural network with a first-order optimizer, retaining the ability to employ state-of-the-art deep learning techniques, such as normalization (Ioffe & Szegedy, 2015; Ba et al., 2016) or dropout (Srivastava et al., 2014). However, this approach has severe drawbacks. For one, the individual optimizations can more easily get stuck in local optima than the combined training (Holl et al., 2020). Also, many inverse problems are inherently multi-modal, i.e. multiple solutions $x^*$ exist for one $y^*$. In such settings $x^{\text{sv}}$ heavily depends on the initial guess $x_0$ used for precomputation, which may cause the network to interpolate between possible solutions, leading to subpar convergence and generalization performance.

To avoid these problems, we alter the training procedure from Eq. 3 in two ways. First, we treat the inverse problem solver $\mathcal{P}^{-1}$ as part of a coupled optimization and consider it as part of the training loop, yielding

$$\theta_* = \arg\min_{\theta} \sum_{i=1}^{N} \frac{1}{2} \|\text{NN}(y_i^* \mid \theta) - \mathcal{P}_{n_i^*}^{-1}(y_i^*)\|_2^2. \tag{4}$$

Next we can condition the embedded inverse problem solver on the neural network prediction, $\mathcal{P}_{n_i^*}^{-1}(y_i^*) \to \mathcal{P}_{n_i^*}^{-1}(y_i^* \mid \text{NN}(y_i^* \mid \theta))$, by using it as an initial guess. The embedded optimizer can now find a minimum close to the prediction, which allows the neural network to choose any solution of a multi-modal problem. Also, since all inverse problems from $\mathcal{Y}^*$ are optimized jointly, this reduces the likelihood that an individual solution gets stuck in a local minimum.

It is furthermore not necessary to run the iterative inverse solver to convergence. Any choice of $n \in \mathbb{N}$ steps likely leads to convergence (proof given in appendix A.2), and choosing small $n$ can

increase the training speed. We also observe this in practice: Optimizations with $n = 1$ converged reliably in all our experiments. With these modifications, we arrive at

$$\theta_* = \arg\min_\theta \sum_{i=1}^N \frac{1}{2} \|\mathrm{NN}(y_i^* \,|\, \theta) - \mathcal{P}_n^{-1}(y_i^* \,|\, \mathrm{NN}(y_i^* \,|\, \theta))\|_2^2. \tag{5}$$

The central quantity here is the difference between prediction and correction obtained from $\mathcal{P}_n^{-1}$, which we refer to as the *physical gradient*. It encodes a valid physical state and takes the place of $\left(\frac{\partial \mathcal{P}}{\partial x}\right)^T$, which would otherwise be computed by backpropagation.

Since the PG stems from a generic higher-order or a domain-specific solver, the resulting updates $\Delta\theta$ will also be non-linear without computing the Hessian w.r.t. $\theta$. Assuming the physics optimizer can account for varying function sensitivity in its updates $\Delta x$, the resulting PG updates $\Delta\theta$ will inherit that property since the network is generally tuned to retain gradient magnitude. The properties of PG as well as GD and Newton's method are summarized in Fig. 2b.

While the $L_2$ measure in Eq. 5 looks like a supervised objective in $x$ space, the actual network optimization does not behave as such. To see this more clearly, consider $\mathcal{P}^{-1}$ performing a single GD step with unit learning rate, $\mathcal{P}^{-1}(y^* \,|\, x) = x - \left(\frac{\partial \mathcal{P}}{\partial x}\right)^T (\mathcal{P}(x) - y^*)$. Then the gradient for Eq. 5 is $\left(\frac{\partial \mathcal{P}}{\partial x} \frac{\partial x}{\partial \theta}\right)^T (\mathcal{P}(x) - y^*)$, which is identical to the unsupervised training in Eq. 2 (the proof is given in appendix A.1).

---

**Algorithm 1:** Neural network training with physical gradients according to Eq. 5

---

**for** *each training sample* **do**

    $x_0 \leftarrow \mathrm{NN}(y^* \,|\, \theta)$                 Solution inference with NN

    **for** $k = 1, ..., n$ **do**

        |   $x_k \leftarrow \mathcal{P}^{-1}(y^* \,|\, x_{k-1})$       Accumulate iterative updates for PG

    **end**

    $\Delta\theta \leftarrow \left(\frac{\partial x}{\partial \theta}\right)^T \cdot (x_0 - x_n)^T$      Backpropagation through NN

    $\theta \leftarrow \mathrm{Step}(\theta, \Delta\theta, \eta)$          Parameter update

**end**

---

Algorithm 1 details the corresponding neural network training procedure. There, the effective gradient $\Delta\theta$ can also be written as $\frac{\partial}{\partial \theta} \frac{1}{2} \|x_0 - x_n\|_2^2$ with $x_n$ taken as constant as in Eq. 5, which lends itself to straightforward integration into machine learning frameworks where a simple $L_2$ loss can be used to optimize the network. This $L_2$ loss acts as a proxy to embed the physical gradient into the network training pipeline and is not to be confused with a traditional supervised loss in $x$ space (see appendix A.1 for a more detailed discussion).

## 3   RESULTS

We first demonstrate the differences between various neural network training schemes on a simple one-parameter example before moving on to physical systems described by partial differential equations. We specifically consider Poisson's equation, the heat equation, and the Navier-Stokes equations. This selection covers ubiquitous physical processes with diffusion, transport, and strongly non-local effects, featuring both explicit and implicit solvers. In each experiment, we train identically-initialized networks using different gradient schemes for $\mathcal{P}$ to examine the relative differences in convergence behavior. For the weight update, we primarily employ the Adam optimizer (Kingma & Ba, 2015), a standard method in deep learning that outperforms SGD in our experiments. All training data are randomly generated on the fly, resulting in effectively infinite data sets $\mathcal{Y}$. All shown learning curves are therefore representative of the performance on unseen data. A detailed description of our experiments along with additional visualizations and performance measurements can be found in appendix B.

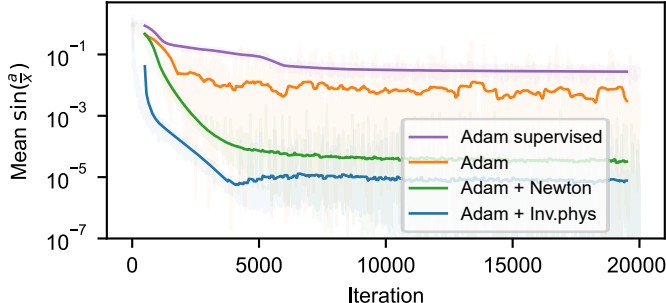

Figure 3: Network trained on single-parameter optimization (section 3.1) using Adam in combination with various gradient schemes. Running average over 1000 mini-batches.

## 3.1 Single-parameter optimization

First, we consider the task of finding solutions $x^* \in \mathbb{R}$ to the inverse problem (Eq. 1) $\mathcal{P}_a(x) = \sin(\frac{a}{x})$, $y^* = -1$, given $a \in [0.1, 10.1]$. This problem may seem simple at first glance but $\mathcal{P}$ shares some properties with chaotic physical systems, which makes this problem hard to optimize. We train a neural network with three fully-connected hidden layers to solve this task; the learning curves are shown in Fig. 3.

Supervised training schemes (Eq. 3) perform poorly on this task due to multi-modality. Using Adam for the joint optimization – with backpropagation through the network and physics (Eq. 2) – avoids the aforementioned problem as the gradients dynamically guide the optimization towards a proximal minimum. However, $\frac{\partial \mathcal{P}}{\partial x}$ oscillates without bound when $x \to 0$, which causes overflow errors in the optimization. To avoid this, we clip the gradients $\frac{\partial \mathcal{P}}{\partial x}$ to $[-1, 1]$, which keeps the optimization relatively stable, although in most cases it never fully converges. This demonstrates one of the key weaknesses of first-order learning: when the magnitude of the gradients varies strongly between examples, the optimization is either slow or unstable, depending on the learning rate.

Finally, we test two variants of physical gradients (Eq. 5) on this problem: Newton's method and an analytic solver. As Fig. 3 shows, the PG methods vastly outperform both supervised and unsupervised first-order training, converging to an accuracy of around $10^{-5}$. For this example, using PG with the inverted physics results in a 1000x improvement in accuracy over conventional network training, while being more stable. The gradients of all variants are visualized in appendix B.2.

## 3.2 Poisson's equation

Poisson's equation, $\mathcal{P}(x) = \nabla^{-2}x$, plays an important role in electrostatics, Newtonian gravity, and fluid dynamics (Ames, 2014). It has the property that local changes in $x$ can affect $\mathcal{P}(x)$ globally. Here we consider a two-dimensional system and train a U-net (Ronneberger et al., 2015) to solve inverse problems (Eq. 1) on pseudo-randomly generated $y^*$. Fig. 4c shows the learning curves.

We construct PGs based on the analytic inverse and couple them with Adam for training the neural network. When learning with Adam or SGD with momentum, learning drastically slows after around 200 and 300 iterations, respectively. Meanwhile, the learning curve involving PGs closely resembles an exponential curve, which indicates linear convergence, the ideal case for first-order methods optimizing an $L_2$ objective. During all of training, the PG variant converges exponentially faster than the first-order alternatives, its relative performance difference compared to Adam continually increasing from a factor of 3 at iteration 60 to a full order of magnitude after 5k iterations.

## 3.3 Heat equation

Next, we consider a system with fundamentally non-invertible dynamics. The heat equation, $\frac{\partial u}{\partial t} = \nu \cdot \nabla^2 u$, models heat flow in solids but also plays a part in many diffusive systems (Droniou, 2014). It gradually destroys information as the temperature equilibrium is approached (Grayson, 1987), causing $\nabla \mathcal{P}$ to become near-singular. Inspired by heat conduction in microprocessors, we generate examples $x_{\text{GT}}$ by randomly scattering four to ten heat generating rectangular regions on a plane and

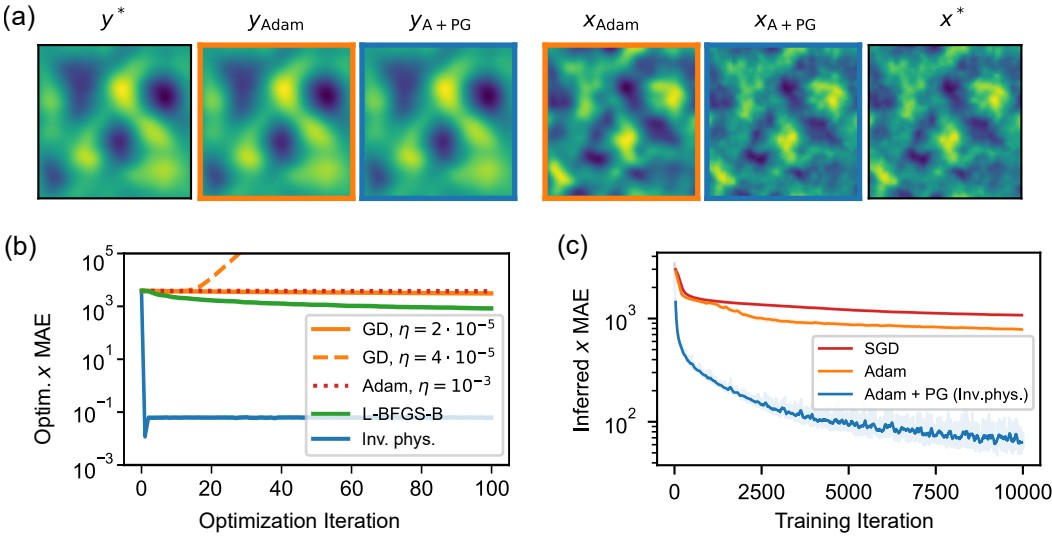

Figure 4: Optimization of Poisson's equation to reconstruct a desired output state (section 3.2). (a) Example from the data set: observed distribution ($y^*$) and inferred solutions, ground truth solution ($x^*$). (b) Convergence curves of various optimization methods on a random example. (c) Neural network learning curves, running average over 64 mini-batches. The vertical axes are logarithmic.

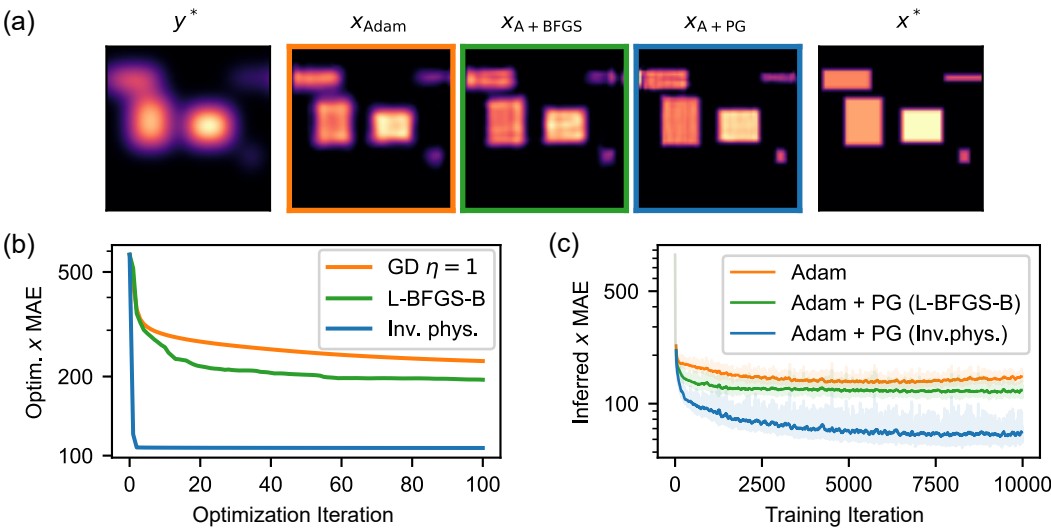

Figure 5: Optimization involving the heat equation (section 3.3). (a) Example from the data set: observed distribution ($y^*$), inferred solutions, ground truth solution ($x^*$). (b) Optimization curves for one example. (c) Learning curves.

simulating the heat profile $y^* = \mathcal{P}(x_{\text{GT}})$ as observed from outside a heat-conducting casing. We train a U-net (Ronneberger et al., 2015) to solve the corresponding inverse problem (Eq. 1). The learning curves are shown in Fig. 5c.

Since $\frac{\partial \mathcal{P}}{\partial x}$ is stable, individual inverse problems can be optimized with GD (see Fig. 5b). For the unsupervised network training (Eq. 2), we use Adam with $\eta = 10^{-3}$ and observe that the distance to the solution starts rapidly decreasing before decelerating between iterations 30 and 40 to a slow but mostly stable convergence. The sudden deceleration is rooted in the adjoint problem, which is also a diffusion problem. Backpropagation through $\mathcal{P}$ removes detail from the gradients, which makes it hard for first-order methods to recover the solution.

For training with PGs (Eq. 5), we consider L-BFGS-B and the analytically derived inverse-physics solver $\mathcal{P}^{-1}$. L-BFGS-B outperforms GD when optimizing single inverse problems but is limited by the ill-conditioning (see Fig. 5b) of the problem. Instead, we use the available domain knowledge to formulate an inverse problem solver $\mathcal{P}^{-1}$ that finds approximate $x^*$ in a stable manner by introducing probabilistic reasoning into the algorithm (see appendix B.4).

Using L-BFGS-B with $n = 32$ as a PG to train the network leads to solutions that are about 20% more accurate and noticeably sharper after the same number of training iterations. This advantage is achieved early during training and stays relatively constant throughout. From iteration 40 on, L-BFGS-B training converges about as fast as pure first-order training. However, the 17x longer computation time per iteration largely negates these advantages.

The PGs based on the inverse-physics solver, on the other hand, manage to improve the convergence speed substantially. They perform on par with L-BFGS-B training for the first ∼35 iterations. However, afterwards the increase in accuracy does not slow down as much as with the other methods, resulting in an exponentially faster convergence. At iteration 100, the predictions are around 34% more accurate compared to Adam, and the difference increases to 130% after 10k iterations. To reach the accuracy of regular Adam training with 10k iterations, L-BFGS-B training requires 628 iterations and training with the inverse-physics solver requires only 138 iterations.

Additional examples and learning curves are shown in section appendix B.4.

### 3.4 NAVIER-STOKES EQUATIONS

Fluids and turbulence are among the most challenging and least understood areas of physics due to their highly nonlinear behavior and chaotic nature (Galdi, 2011). We consider a two-dimensional system governed by the incompressible Navier-Stokes equations: $\frac{\partial v}{\partial t} = \nu \nabla^2 v - \nabla \cdot \nabla v - \nabla p$, $\nabla \cdot v = 0$, $\nabla \times p = 0$, where $p$ denotes pressure and $\nu$ the viscosity. At $t = 0$, a region of the fluid is randomly marked with a massless colorant $m_0$ that passively moves with the fluid, $\frac{\partial m}{\partial t} = -v \cdot \nabla m$, and after a time $t$, the marker is observed again, $m_t$. An example observation pair $y^* = \{m_0, m_t\}$ is shown in Fig. 6a. We target the inverse problem (Eq. 1) of finding an initial fluid velocity $x \equiv v_0$ such that the fluid simulation $\mathcal{P}$ matches $m_t$ at time $t$. Since $\mathcal{P}$ is deterministic, $x$ encodes the complete fluid flow from $0$ to $t$. We define the objective in frequency space with lower frequencies being weighted more strongly. This definition considers the marker distribution match on all scales, from the coarse global match to fine details, and is compatible with the definition in Eq. 1. We train a U-net (Ronneberger et al., 2015) to solve these inverse problems; the learning curves are shown in Fig. 6c.

For unsupervised training (Eq. 2) with Adam, the error decreases for the first 100 iterations while the network learns to infer velocities that lead to an approximate match. The error then proceeds to decline at a much lower rate, nearly coming to a standstill. This is caused by an overshoot in terms of vorticity, as visible in Fig. 6a right. While the resulting dynamics can roughly approximate the shape of the observed $m_t$, they fail to match its detailed structure. Moving from this local optimum to the global optimum is very hard for the network as the distance in $x$ space is large and the gradients become very noisy due to the highly non-linear physics. A similar behavior can also be seen when optimizing single inverse problems with GD (Fig. 6b). Here the strongly varying magnitude of the gradients poses an additional challenge for the first-order optimization, causing its progress to stall. With $\eta = 1$, it takes more than 20K iterations for GD to converge for single problems.

For training with PGs (Eq. 5), we exploit our knowledge about the flow – e.g. smoothness – to formulate an inverse simulator $\mathcal{P}^{-1}$ that incorporates corresponding priors on the solution space. It runs the simulation in reverse, starting with $m_t$, then estimates $x$ by comparing the marker densities from the forward and reverse pass (see appendix B.5 for details). This method converges for single inverse problems within around 100 iterations and its computational costs are on the same level as evaluating the gradient. When used for network training, we observe vastly improved convergence behavior compared to conventional first-order training. The error rapidly decreases during the first 200 iterations, at which point the inferred solutions are more accurate than pure Adam training by a factor of 2.3. The error then continues to improve at a slower but still exponentially faster rate, reaching a relative accuracy advantage of 5x after 20K iterations. To match the network trained with pure Adam for 20K iterations, the PG variant only requires 55 iterations. This improvement is possible because the inverse-physics solver and associated PG does not suffer from the strongly-varying

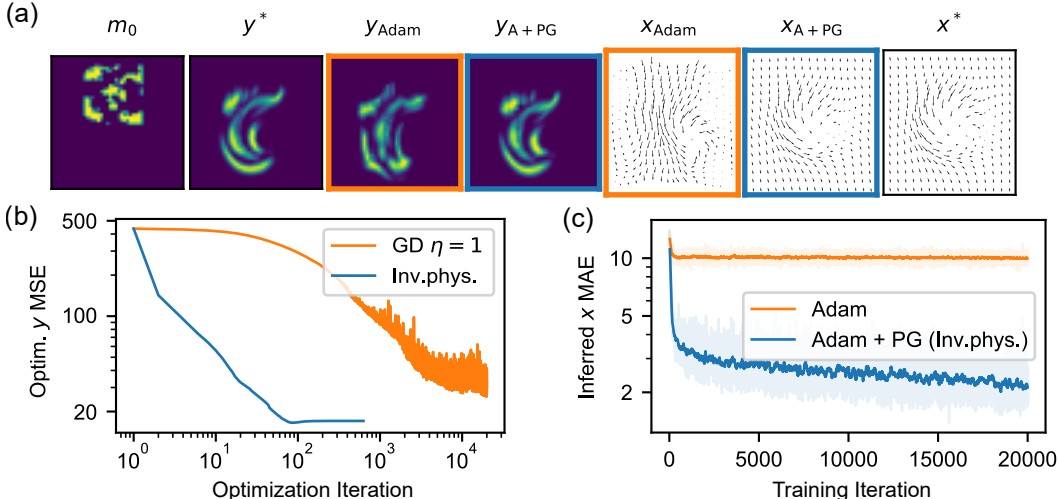

Figure 6: Incompressible fluid flow (section 3.4). (a) One example from the data set: initial marker distribution ($m_0$); simulated marker distribution after time $t$ using ground-truth velocity ($y^*$) and network predictions ($y$.); predicted initial velocities ($x$.); ground truth velocity ($x^*$). (b) Optimization curves with $\eta = 1$, averaged over 4 examples. (c) Learning curves, running average over 64 mini-batches.

Table 1: Time to reach equal solution quality in the fluid experiment, measured as MAE in $x$

| Method | Training time | Inference time per example |
| --- | --- | --- |
| Neural network with PG | 17.6 h (15.6 k iterations) | 0.11 ms (immediate) |
| Domain-specific solver | n/a | 2.2 s (7 iterations) |
| Gradient descent optimizer | n/a | > 4h (20k iterations) |

gradient strengths and directions, which drown the first-order signal in noise. Instead, physical gradients behave much more smoothly, both in magnitude and direction.

To reach the same solution quality as the neural network prediction, the domain knowledge solver needs more than 10,000 times as long. This difference is caused by solver having to run the full forward and backward simulation several times. Table 1 lists measured training and inference times. With both iterative solver and network, we used a batch size of 64 and divided the total time by the batch size. More examples from the fluid data set including time evolution sequences are shown in appendix B.5.

## 4 CONCLUSION

We have shown how to integrate arbitrary optimization schemes for inverse problems into machine learning settings. Our results show that first-order optimization is suboptimal in many situations, especially when the magnitude or direction of gradients can vary during the optimization or from example to example. While more advanced optimizers such as Adam alleviate this behavior, they fail to determine good optimization directions for many problems. By embedding higher-order optimizers – and especially inverse-physics solvers – into the first-order training pipeline, we were able to mitigate these shortcomings without discarding the progress made in the field of machine learning. The resulting training scheme converges exponentially faster than traditional first-order training on a wide variety of inverse problems involving partial differential equations while being on par with regular gradient evaluation in terms of computational cost.

We anticipate our method to be applicable to a wide range of differentiable processes. Beyond physics, interesting avenues lie in applying PGs to differentiable rendering or training invertible neural networks.

## REPRODUCIBILITY STATEMENT

We will publish the full source code and trained network models upon acceptance. They will be sufficient to retrain the networks and plot the figures shown in this work. All data sets were generated randomly, so slight variations are expected. Repeated experiments are shown in the appendix.

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

## A   METHOD

### A.1   EMBEDDING THE PHYSICAL GRADIENT INTO THE GRADIENT DESCENT PIPELINE

The key for embedding inverse-physics optimizers in the machine learning pipeline is the proxy $L_2$ loss in Eq. 5 which passes the gradients from the physics optimizer to the GD-based neural network optimizer.

This $l_2$ loss may look like a supervised loss in $x$ space but due to the dependency of the labels $\mathcal{P}^{-1}$ on the prediction, it behaves very differently. To demonstrate this, we consider the case that GD is being used as the physics optimizer. Then the total loss is purely defined in $y$ space, reducing to a regular first-order optimization. The proxy $L_2$ loss simply connects the computational graphs used in backpropagation.

**Physical gradients based on gradient descent are equal to unsupervised training**   With the shorthands $x_i = \mathrm{NN}(y_i^* \,|\, \theta)$, $y_i = \mathcal{P}(x_i)$ and $\Delta y_i = y_i - y_i^*$, we can write the objective functions of unsupervised training and physical gradient training as

$$U(\theta) = \sum_{i=1}^{N} \frac{1}{2} ||\Delta y_i||_2^2 \quad \text{(unsupervised)} \tag{6}$$

$$M(\theta) = \sum_{i=1}^{N} \frac{1}{2} ||x_i - \mathcal{P}^{-1}(y_i^* \,|\, x_i)||_2^2 \quad \text{(physical gradients)} \tag{7}$$

**Theorem 1.** *Minimizing $U(\theta)$ is identical to minimizing $M(\theta)$ using any gradient-descent-based optimizer if $\mathcal{P}^{-1}$ is implemented as a single gradient descent step with $\eta = 1$.*

*Proof.* The gradient of the unsupervised objective function is

$$\frac{\partial U}{\partial \theta} = \sum_{i=1}^{N} \Delta y_i \cdot \frac{\partial y}{\partial x} \frac{\partial x}{\partial \theta}$$

For the mixed training, we recall that the objective function of a single inverse problem is $L(x) = \frac{1}{2} ||\mathcal{P}(x) - y^*||_2^2$. The corresponding gradient is $\frac{\partial L}{\partial x} = \Delta y \cdot \frac{\partial y}{\partial x}$. Inserting a gradient descent step for $\mathcal{P}^{-1}$, the gradient of the mixed objective function becomes

$$\frac{\partial M}{\partial \theta} = \sum_{i=1}^{N} \left( x_i - \mathcal{P}^{-1}(y_i^* \,|\, x_i) \right) \frac{\partial x}{\partial \theta}$$

$$= \sum_{i=1}^{N} \left( x_i - \left( x_i - \eta \cdot \frac{\partial L}{\partial x} \right) \right) \frac{\partial x}{\partial \theta}$$

$$= \sum_{i=1}^{N} \eta \cdot \Delta y_i \cdot \frac{\partial y}{\partial x} \frac{\partial x}{\partial \theta}$$

which is equal to $\frac{\partial U}{\partial \theta}$ for $\eta = 1$. $\qquad\qquad\square$

This generic $L_2$ formulation is even applicable in settings where different parts of an end-to-end pipeline are computed by different software frameworks or hardware accelerators.

### A.2   CONVERGENCE WITH PHYSICAL GRADIENTS

We consider the convergence of a coupled optimization of $L_i(x_i)$ with $x_i = f_\theta(y_i)$ involving gradient descent optimization of a general function approximator $f_\theta$ and some other optimization scheme $P$ for minimizing objective functions $L_i(x_i)$. Here, $i$ denotes the index of a specific example $y$ from a data set of size $N$. In the context of learning to solve inverse problems, $P$ is the inverse-problem solver on which physical gradient $P(x) - x$ is based, and $f_\theta$ represents the neural network. In the main text, we considered objectives of the form $L_i = \frac{1}{2} ||\mathcal{P}(x_i) - y_i^*||_2^2$, but now we consider a more general form.

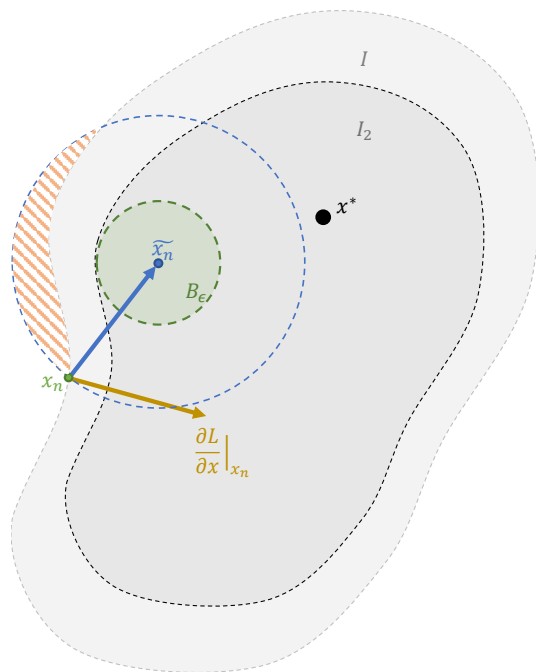

Figure 7: Convergence visualization in $x$ space for one example $i$. The point $x_n = f_\theta(y)$ represents the current solution estimate. The grey line and area mark all $\{x | L(x) < L(x_n)\}$ and $x^*$ is a minimum of $L$. $\tilde{x}_n = P(x_n)$ is the next point in the optimization trajectory of $x$ and the green and blue circles around it represent open sets with radii $\epsilon$ and $||\tilde{x}_n - x_n||_2$, respectively. In the area shaded in orange, the distance to $\tilde{x}_n$ decreases but $L$ increases.

**Assumptions**

1. Let $L = \{L_i : \mathbb{R}^d \to \mathbb{R} \,|\, i = 1, ..., N\}$ be a finite set of functions and let $x_i^* \in \mathbb{R}^d$ be any global minima of $L_i$.

2. Let $P = \{P_i : \mathbb{R}^d \to \mathbb{R}^d \,|\, i = 1, ..., N\}$ be a set of update functions for $L_i$ such that $\exists \tau > 0 : \forall x \in \mathbb{R}^d : L_i(x) - L_i(P_i(x)) \geq \tau (L(x) - L(x^*))$ and $\exists K > 0 : \forall x \in \mathbb{R}^d : ||P_i(x) - x|| \leq K(L(x) - L(x^*))$.

3. Let $f_\theta : \mathbb{R}^m \to \mathbb{R}^d$ be differentiable w.r.t. $\theta$ with the property that $\exists \eta > 0 : \forall i \in 1, ..., N \; \forall x_i \in \mathbb{R}^d \; \forall \epsilon > 0 \; \exists n \in \mathbb{N} : ||f_{\theta_n^{x_i}}(y_i) - x_i||_2 \leq \epsilon$ where $\theta_n^x$ is the sequence of gradient descent steps with $\theta_{n+1}^x = \theta_n^x - \eta \left(\frac{\partial f_\theta}{\partial \theta}\right)^T (f_{\theta_n} - x)$, $\eta > 0$.

Here we make minimal assumptions about $L$; not even continuity is required as long as we have access to some form of a physical gradient that can optimize it in $x$. We denote $k$ repeated applications of $P$ by $P^k(x)$. The assumption in (3) states that the function $f_\theta$ is flexible enough to fit our problem. It must be able to converge to every point $x$ for all examples using gradient descent. It has been shown that neural networks with sufficiently many parameters fulfill this condition under certain assumptions (Du et al., 2018) and the universal approximation theorem guarantees that such a configuration exists even for shallow neural networks with enough parameters (Cybenko, 1989).

We now show that $P$ can be used in combination with gradient descent to optimize the coupled problem.

**Theorem 2.** *For all $k \in \mathbb{N}$, there exists an update strategy $\theta_{n+1} = U_k(\theta_n)$ based on a single evaluation of $P^k$ for which $L(f_{\theta_n}(y_i))$ converges to a minimum $x_i^*$ or minimum region of $L_i$ $\forall i = 1, ..., N$.*

*Proof.* Dropping the example index $i$, we denote $\tilde{x}_n \equiv P^k(x_n)$ and $\Delta L^* = L(x_n) - L(x^*)$. Let $I_2$ denote the open set of all $x$ for which $L(x) - L(x_n) > \frac{\tau}{2}\Delta L^*$ Definition (2) provides that $\tilde{x}_n \in I_2$.

Since $I_2$ is open, $\exists \epsilon > 0 : \forall x \in B_\epsilon(\tilde{x}_n) : L(x_n) - L(x) > \frac{\tau}{2}\Delta L^*$ where $B_\epsilon(x)$ denotes the open set containing all $x' \in \mathbb{R}^d$ for which $||x' - x||_2 < \epsilon$, i.e. there exists a small ball around $\tilde{x}_n$ which is fully contained in $I_2$ (see sketch in Fig. 7).

Using the convergence assumption for $f_\theta$, definition (3), we can find a finite $n \in \mathbb{N}$ for which $f_{\theta_n} \in B_\epsilon(\tilde{x}_n)$ and therefore $L(x_n) - L(f_{\theta_n}) > \frac{\tau}{2}\Delta L^*$. We can thus use the following strategy $U_k$ for minimizing $L(f_\theta)$: First, compute $\tilde{x}_n = P^k(x_n)$. Then perform gradient descent steps in $\theta$ with the effective objective function $\frac{1}{2}||f_\theta - \tilde{x}_n||_2^2$ until $L(x_n) - L(f_\theta) \geq \frac{\tau}{2}\Delta L^*$. With this strategy, each application of $U_k$ reduces the loss to $\Delta L_{n+1}^* \leq (1 - \frac{\tau}{2})\Delta L_n^*$ so any arbitrary value of $L > L(x^*)$ can be reached within a finite number of steps. Definition (2) also ensures that $||P(x) - x|| \to 0$ as the optimization progresses which guarantees that the optimization converges to a minimum region. $\square$

While this guarantees convergence, it requires potentially many gradient descent steps in $\theta$ for each physical gradient evaluation. This can be advantageous in special circumstances, e.g. when the physical gradient is more expensive to compute than an update in $\theta$, and $\theta$ is far away from a solution. However, in many cases, we want to update the physical gradient after each update to $\theta$. Without additional assumptions about $P$ and $f_\theta$, there is no guarantee in this case that $L$ will decrease every iteration, even for infinitesimally small step sizes. Despite this, there is good reason to assume that the optimization decreases $L$ over time.

When performing a gradient descent step in $\theta$ for the objective $\frac{1}{2}||f_\theta - \tilde{x}_n||_2^2$, the next value of $f_\theta$ must lie closer to $\tilde{x}_n$ than $x_n$. Assuming that, on average, the gradient descent steps do not prefer a certain direction, $x_{n+1}$ is more likely to lie in $I = \{x \in \mathbb{R}^d : L(x) < L(x_n)\}$ than outside it. The region of increasing loss is shaded orange in Fig. 7. Since this region always fills less than half of the sphere around $\tilde{x}_n$, $L$ is more likely to decrease than increase.

While this shows that the loss should decrease on average, it does not guarantee convergence. We now look at two specific formulations of $P$ for which convergence to the correct solution is guaranteed. The first involves the physical gradient pointing directly towards a solution $x^*$. This is, for example, the case for unimodal problems when we choose $k$ large enough. In the second case, we consider the physical gradient being aligned with the gradient descent vector in $x$ space.

**Theorem 3.** *If $\forall x \in \mathbb{R}^d \, \exists \lambda \in (0,1] : P(x) = x + \lambda(x^* - x)$, then the sequence $f_{\theta_n}$ with $\theta_{n+1} = \theta_n - \eta \left(\frac{\partial f}{\partial \theta}\right)^T (x - P(x))$ converges to $x^*$.*

*Proof.* Rewriting $x - P(x) = \frac{\partial}{\partial x}\left(\frac{\lambda}{2}||x - x^*||_2^2\right)$ yields the update $\theta_{n+1} - \theta_n = -\eta \left(\frac{\partial \tilde{L}}{\partial x}\frac{\partial f}{\partial \theta}\right)^T$ where $\tilde{L} = \frac{\lambda}{2}||x - x^*||_2^2$. This describes pure gradient descent towards $x^*$ with the gradients scaled by $\lambda$. Since $\lambda \in (0, 1]$, the convergence proof of gradient descent applies. $\square$

**Theorem 4.** *If $\forall x \in \mathbb{R}^d \, \exists \lambda \in (0,1] : P(x) = x - \lambda \left(\frac{\partial L}{\partial x}\right)^T$, then the sequence $f_{\theta_n}$ with $\theta_{n+1} = \theta_n - \eta \left(\frac{\partial f}{\partial \theta}\right)^T (x - P(x))$ converges to minima of $L$.*

*Proof.* This is equivalent to gradient descent in $L(\theta)$. Rewriting the update yields $\theta_{n+1} - \theta_n = -\eta \left(\frac{\partial f}{\partial \theta}\right)^T \left(x - \left(x - \left(\lambda \frac{\partial L}{\partial x}\right)^T\right)\right) = -\eta\lambda \left(\frac{\partial L}{\partial x}\frac{\partial f}{\partial \theta}\right)^T$ which is the gradient descent update scaled by $\lambda$. $\square$

## B    EXPERIMENTS

Here, we give a more detailed description of our experiments including setup and analysis. The implementation of our experiments is based on the $\Phi_{\text{Flow}}$ (PhiFlow) framework (Holl et al., 2020) and uses TensorFlow (Abadi et al., 2016) and PyTorch (Paszke et al., 2017) for automatic differentiation. Our code is open source and will be made available upon acceptance.

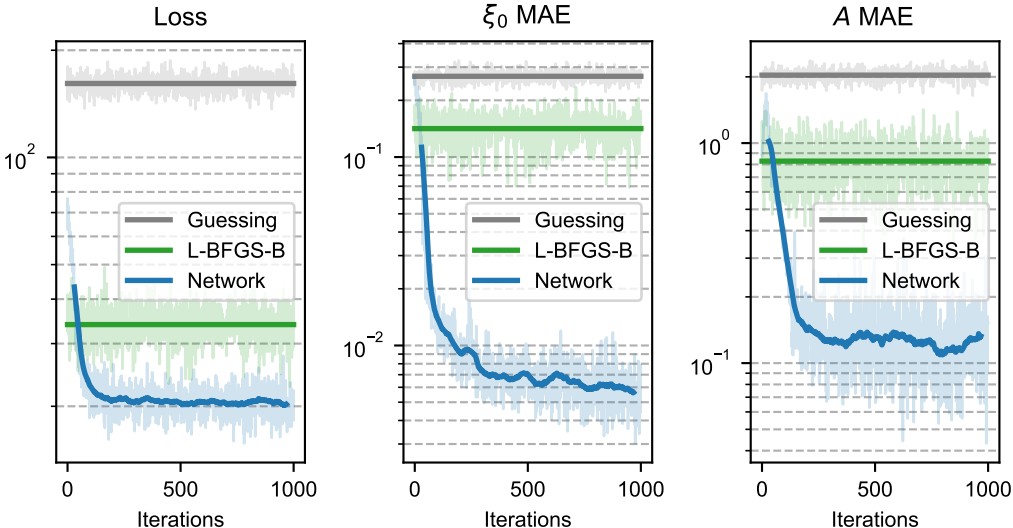

Figure 8: Learning curves of the network trained to fit wave packets. The performance of L-BFGS-B and random guessing are shown on the same data for reference. The left graph shows the objective $||y - y^*||_2^2$ and the right two graphs show the $x$-space deviation from the true solution in position $\xi_0$ and amplitude $A$.

### B.1 WAVE PACKET FIT

This experiment is an instance of a generic curve fitting problem. The task is to find the parameters that result in least mean squared error between two curves. We use this problem to illustrate the advantages of using neural networks as inverse solvers over classical optimization algorithms.

**Data generation.** We simulate an observed time series $y^*$ from a random ground truth position and amplitude $x^* = \{\xi_0, A\}$. Each time series contains 256 entries and is made up of the wave packet and noise. For the wave packet, we sample $\xi_0 \in [25.6, 230.4)$ and $A \in [1, 4)$ from uniform distributions. The wave packet has the functional form

$$y(\xi) = A \cdot \sin(f \cdot \xi) \cdot \exp\left(-\frac{1}{2}\frac{(\xi - \xi_0)^2}{\sigma^2}\right)$$

where we set $f = \frac{4}{5}$ and $\sigma = 20$ constant for all data. For the noise, we superimpose random values sampled from the normal distribution $\mathcal{N}(0, \frac{2}{5})$ at each sample.

**Network architecture.** We construct the neural network from convolutional blocks, followed by fully-connected layers, all using the ReLU activation function. The input is first processed by five blocks, each containing a max pooling operation and two 1D convolutions with kernel size 3. Each convolution outputs 16 feature maps. The downsampled result is then passed to two fully connected layers with 64 and 32 and 2 neurons, respectively, before a third fully-connected layer produces the predicted $\xi_0$ and $A$. $\xi_0$ is passed through a Sigmoid activation function and normalized to values between 25.6 and 230.4.

**Training and fitting.** We fit the data using L-BFGS-B with an initial guess of $\xi_0 = 128$ and $A = 2$ and the network output is offset by the same amount. Both network and L-BFGS-B minimize the squared loss $||y(\xi_0, A) - y^*||_2^2$ and the resulting performance curves are shown in Fig. 8. We observe that L-BFGS-B manages to fit the wave packet when it is reasonably close to the center where the initial guess predicts it. When the wave packet is located to either side, L-BFGS-B does not find it and instead fits the noise near the center.

The neural network is trained using Adam with learning rate 0.001 and batch size of 100. Despite the simpler, Hessian-free, optimization updates, the network quickly learns to localize all wave packets,

outperforming L-BFGS-B after 30 to 40 training iterations. This improvement is possible because of the network's reparameterization of the problem, allowing for joint parameter optimization using all data. When the prediction for one example is close to a local optimum, updates from different examples can prevent it from converging to that sub-optimal solution.

## B.2 SINGLE-PARAMETER OPTIMIZATION

The optimization consists of finding $x^* = \text{argmin}_x \sin(\frac{a}{x})$. Fig. 9 shows the gradients for the case $a = 1$. The PG as well as the supervised methods require finding a solution $x^*$ closest to an initial guess $x_0$. We solve this analytically by first determining the index of the closest minimum, $n = \frac{a}{2\pi \cdot x_0} - \frac{3}{4}$ which we round to the closest integer. The furthest-out minima ($n = 0$ and $n = -1$) require special handling. Then the position of the minimum corresponding to $n$ is $x^* = \frac{a}{3\pi/2 + 2\pi \cdot n}$.

**Newton's method** The analytic form of a Newton update is

$$\mathcal{P}(x) - x = -\frac{x^2 \cos(a/x)}{2x \cos(a/x) - a \sin(a/x)}$$

which approaches both minima and maxima. We alter this update to walk only towards minimum points by flipping the optimization direction so that the update points in the same direction as $\frac{\partial \mathcal{P}}{\partial x}$. Since Newton's method relies on an inversion of the Hessian, the produced physical gradients diverge where $\frac{\partial^2 \mathcal{P}}{\partial x^2} = 0$. To avoid overflow, we clip the gradients $\frac{\partial \mathcal{P}}{\partial x}$ to $[-1, 1]$ before they are backpropagated through the neural network.

**Inverse gradients** In addition to gradient inversion via the Hessian, we can invert the gradients directly. The principle of inverting the Jacobian is not new. Inverse kinematics applications such as controlling robots also employ first-order inversion (Craig, 2005). This results in a similar behavior concerning dimensions and function sensitivity as Newton-type methods. However, this first-order inversion cannot directly be applied to general optimization tasks because the Jacobian of a scalar objective function is a row vector. For this specific inverse problem, however, the gradient can be inverted because only a single parameter is optimized. The physical gradient is computed as

$$\mathcal{P}(x) - x = -\frac{h}{\nabla \mathcal{P}} = -x^2 \frac{\sin(a/x) + 1}{a \cos(a/x)}$$

where $h = \sin\left(\frac{a}{x}\right) + 1$ measures the height above $y^*$. It approaches $\pm\infty$ when $\sin(\frac{a}{x})$ is maximal. This inversion requires domain knowledge since $h$ is based on the fact that $\min \mathcal{P} = -1$. Like with Newton's method, we apply gradient clipping to bound diverging inverse gradients.

**Neural network training** We set up a multilayer perceptron with a single input and output as well as three hidden layers containing 10, 20 and 10 neurons, respectively. This adds up to 461 total trainable parameters. A bias is applied at each layer and the ReLU activation function is applied after each hidden layer. The network is trained using Adam with a learning rate of 0.002 and mini-batches containing 1000 uniformly sampled $a \in [0.1, 10.1]$. We use $\Phi_{\text{Flow}}$ together with PyTorch to train the neural network in this experiment.

Learning curves for different network initializations are shown in Fig. 10 and the recorded computation times are shown in Fig. 14. While supervised training shows very consistent behavior across training runs, the convergence curves of Adam show more erratic behavior. Training with PGs based on the analytic solver converges in all cases, but does not always reach the same level of accuracy. While yielding superior accuracy than pure Adam for the large majority of tested initializations, it sometimes stops converging to an accuracy larger than $10^{-4}$. This is most likely caused by numerical effects of the inverse solver. Training with physical gradients based on Newton's method or inverse gradients yields a convergence that strongly depends on the network initialization. This variant converges only for certain initializations. However, upon convergence, it typically reaches an accuracy similar to the inverse physics variant.

## B.3 POISSON'S EQUATION

We consider Poisson's equation, $\nabla^2 y = x$ where $x$ is the initial state and $y$ is the output of the simulator. We set up a two-dimensional simulation with 80 by 60 cubic cells. Our simulator computes

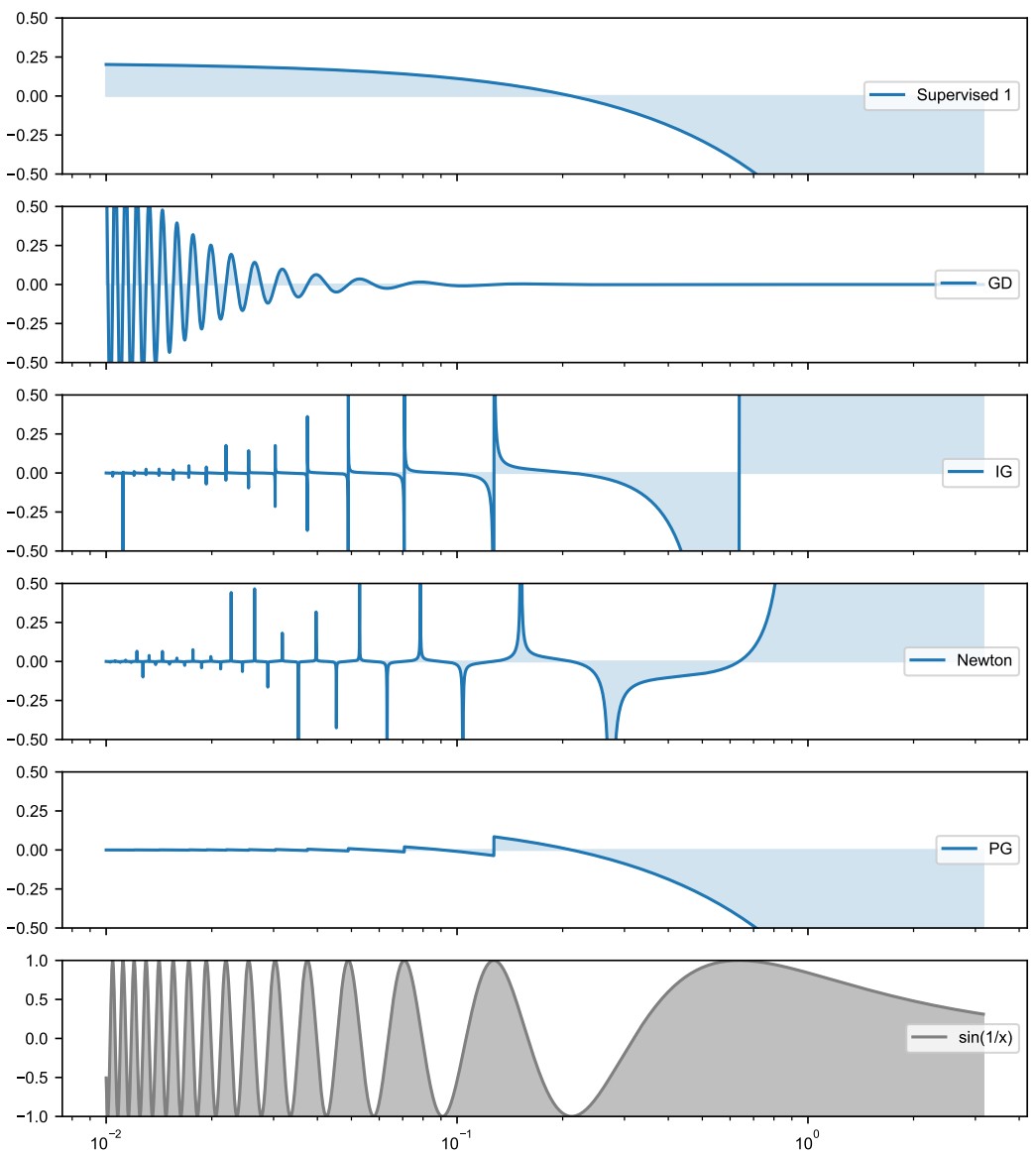

Figure 9: Gradients for minimizing $\sin(a/x)$, shown for $a = 1$. From top to bottom: Supervised learning with $x_0 = 1$, Gradient descent, inverse gradient, Newton's method, physical gradient, objective function. Singularities for small $x$ are not properly resolved in the IG / Newton plots.

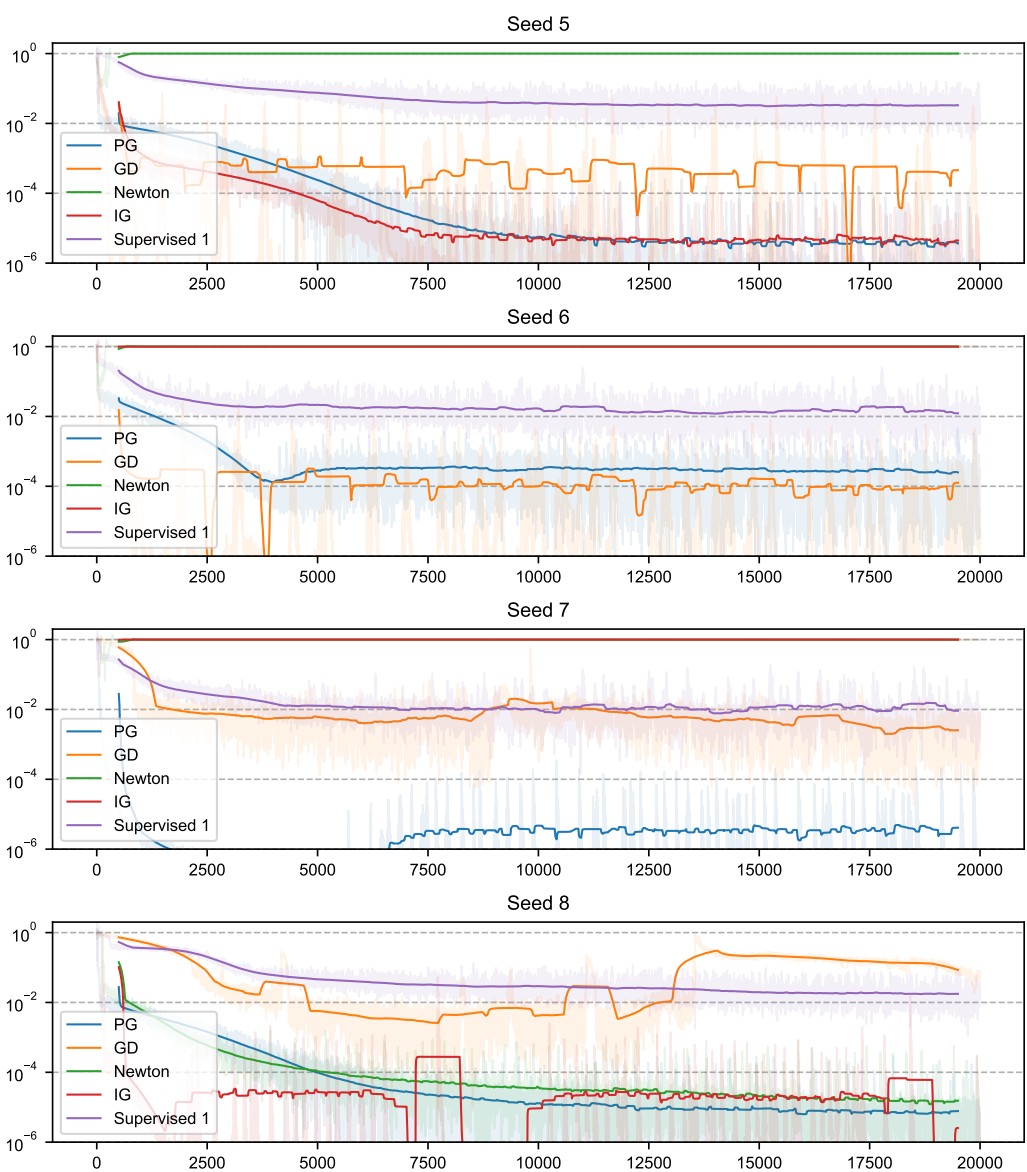

Figure 10: Networks trained on single-parameter optimization using Adam and various gradient schemes. Each figure shows the learning curves for a network initialized with a fixed seed between 5 and 8. The x axis denotes the number of training iterations, Solid lines show the running average over 1000 mini-batches.

$y = \mathcal{P}(x) = \nabla^{-2}x$ implicitly via the conjugate gradient method. The inverse problem consists of finding an initial value $x^*$ for a given target $y^*$ such that $\nabla^2 y^* = x^*$. We formulate this problem as minimizing $L(x) = \frac{1}{2}||\mathcal{P}(x) - y^*||_2^2 = \frac{1}{2}||\nabla^{-2}(x - x^*)||_2^2$. We now investigate the computed updates $\Delta x$ of various optimization methods for this problem.

**Gradient descent** Gradient descent prescribes the update $\Delta x = -\eta \cdot \left(\frac{\partial L}{\partial x}\right)^T = -\eta \cdot \nabla^{-2}(y - y^*)$ which requires an additional implicit solve for each optimization step. This backward solve produces much larger values than the forward solve, causing GD-based methods to diverge from oscillations unless $\eta$ is very small. We found that GD requires $\eta \leq 2 \cdot 10^{-5}$, while the momentum in Adam allows for larger $\eta$. For both GD and Adam, the optimization converges extremely slowly, making GD-based methods unfeasible for this problem.

**Physical gradients via analytic inversion** Poisson's equation can easily be inverted analytically, yielding $x = \nabla^2 y$. Correspondingly, we formulate the update step as $\Delta x = -\eta \cdot \frac{\partial x}{\partial y} \cdot (y - y^*) = -\eta \cdot \nabla^2 (y - y^*)$ which directly points to $x^*$ for $\eta = 1$. Here the Laplace operator appears in the computation of the optimization direction. This is much easier to compute numerically than the Poisson operator used by gradient descent. Consequently, no additional implicit solve is required for the optimization and the cost per iteration is less than with gradient descent. This computational advantages also carries over to neural network training where this method can be integrated into the backpropagation pipeline as a physical gradient.

**Neural network training** We first generate ground truth solutions $x^*$ by adding fluctuations of varying frequencies with random amplitudes. From these $x^*$, we compute $y^* = \mathcal{P}(x^*)$ to form the set of target states $\mathcal{Y}$. Both generation of $x^*$ and $y^*$ is performed on the fly, resulting in a data set of effectively infinite size, $|\mathcal{Y}| = \infty$. This has the advantage that learning curves are representative of both test performance as well as training performance. The top of Fig. 11 shows some examples generated this way. We train a U-net (Ronneberger et al., 2015) with a total of 4 resolution levels and skip connections. The network receives the feature map $y^*$ as input. Max pooling is used for downsampling and bilinear interpolation for upsampling. After each downsampling or upsampling operation, two blocks consisting of 2D convolution with kernel size of 3x3, batch normalization and ReLU activation are performed. All of these convolutions output 16 feature maps and a final 1x1 convolution brings the output down to one feature map. The network contains a total of 37,697 trainable parameters.

For SGD and Adam training, the composite gradient of NN $\circ$ $\mathcal{P}$ is computed with TensorFlow or PyTorch, enabling an end-to-end optimization. The learning rate is set to $\eta = 10^{-3}$ with Adam and $\eta = 10^{-9}$ for SGD. The extremely small learning rate for SGD is required to balance out the large gradients and is consistent with the behavior of gradient-descent optimization on single examples where an $\eta = 2 \cdot 10^{-5}$ was required. We use a typical value of 0.9 for the momentum of SGD and Adam. For the training using Adam with physical gradients, we compute $\Delta x$ as described above and keep $\eta = 10^{-3}$. For each case, we set the learning rate to the maximum value that consistently converges. The learning curves for three additional random network initializations are shown at the bottom of Fig. 11, while Fig. 14 shows the computation time per iteration.

## B.4 HEAT EQUATION

We consider a two-dimensional system governed by the heat equation $\frac{\partial u}{\partial t} = \nu \cdot \nabla^2 u$. Given an initial state $x = u_0$ at $t_0$, the simulator computes the state at a later time $t_*$ via $y = u_* = \mathcal{P}(x)$. Exactly inverting this system is only possible for $t \cdot \nu = 0$ and becomes increasingly unstable for larger $t \cdot \nu$ because initially distinct heat levels even out over time, drowning the original information in noise. Hence the Jacobian of the physics $\frac{\partial y}{\partial x}$ is near-singular. In our experiment we set $t \cdot \nu = 8$ on a domain consisting of 64x64 cells of unit length. This level of diffusion is challenging, and diffuses most details while leaving the large-scale structure intact.

We apply periodic boundary conditions and compute the result in frequency space where the physics can be computed analytically as $\hat{y} = \hat{x} \cdot e^{-k^2(t_* - t_0)}$ where $\hat{y}_k \equiv \mathcal{F}(y)_k$ denotes the $k$-th element of the Fourier-transformed vector $y$. Here, high frequencies are dampened exponentially. The inverse

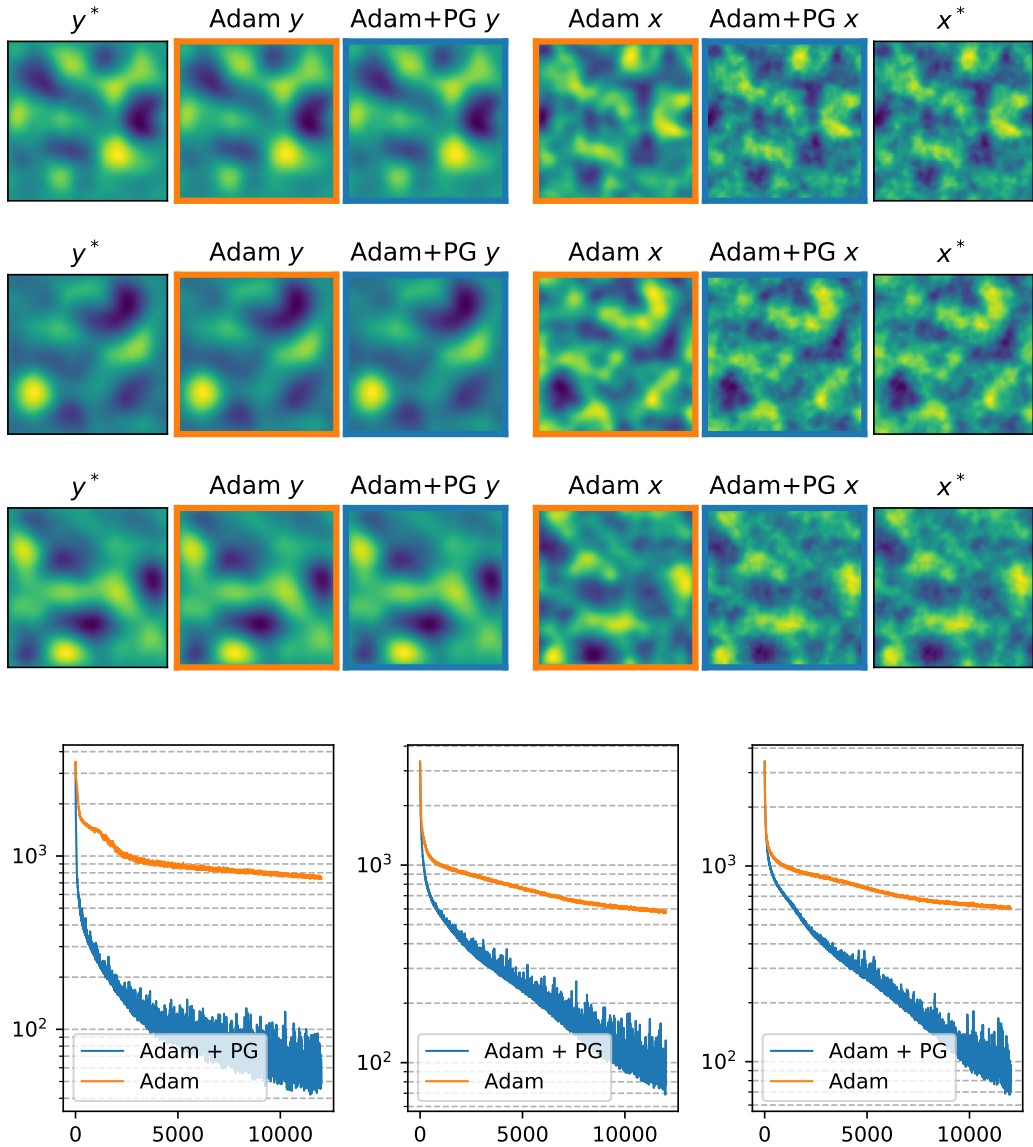

Figure 11: Inverse problems involving Poisson's equation. **Top:** Three examples from the data set, from left to right: observed target ($y^*$), simulated observations resulting from network predictions (Adam $y$, Adam+PG $y$), predicted solutions (Adam $x$, Adam+PG $x$), ground truth solution ($x^*$). Networks were trained for 12k iterations. **Bottom:** Neural network learning curves for three random network initializations, measured as $||x - x^*||_1$.

problem can thus be written as minimizing $L(x) = ||\mathcal{P}(x) - y^*||_2^2 = ||\mathcal{F}^{-1}\left(\mathcal{F}(x) \cdot e^{-k^2(t_* - t_0)}\right) - y^*||_2^2$

**Gradient descent**   Using the analytic formulation, we can compute the gradient descent update as

$$\Delta x = -\eta \cdot \mathcal{F}^{-1}\left(e^{-k^2(t_* - t_0)}\mathcal{F}(y - y^*)\right).$$

GD applies the forward physics to the gradient vector itself, which results in updates that are stable but lack high frequency spatial information. Consequently, GD-based optimization methods converge slowly on this task after fitting the coarse structure and have severe problems in recovering high-frequency details. This is not because the information is fundamentally missing but because GD cannot adequately process high-frequency details.

**Stable physical gradients**   The frequency formulation of the heat equation can be inverted analytically, yielding $\hat{x}_k = \hat{y}_k \cdot e^{k^2(t_* - t_0)}$. This allows us to define the update

$$\Delta x = -\eta \cdot \mathcal{F}^{-1}\left(e^{k^2(t_* - t_0)}\mathcal{F}(y - y^*)\right).$$

Here, high frequencies are multiplied by exponentially large factors, resulting in numerical instabilities. When applying this formula directly to the gradients, it can lead to large oscillations in $\Delta x$. This is the opposite behavior compared to Poisson's equation where the GD updates were unstable and the PG stable.

The numerical instabilities here can, however, be avoided by taking a probabilistic viewpoint. The observed values $y$ contain a certain amount of noise $n$, with the remainder constituting the signal $s = y - n$. For the noise, we assume a normal distribution $n \sim \mathcal{N}(0, \epsilon \cdot y)$ with $\epsilon > 0$ and for the signal, we assume that it arises from reasonable values of $x$ so that $y \sim \mathcal{N}(0, \delta \cdot e^{-k^2})$ with $\delta > 0$. With this, we can estimate the probability of an observed value arising from the signal using Bayes' theorem $p(s|v) = \frac{p(v|s) \cdot p(s)}{p(v|s) \cdot p(s) + p(v|n) \cdot p(n)}$ where we assume the priors $p(s) = p(n) = \frac{1}{2}$. Based on this probability, we dampen the amplification of the inverse physics which yields a stable inverse. Gradients computed in this way hold as much high-frequency information as can be extracted given the noise that is present. This leads to a much faster convergence and more precise solution than any generic optimization method.

**Neural network training**   For training, we generate $x^*$ by randomly placing between 4 and 10 hot rectangles of random size and shape in the domain and computing $y = \mathcal{P}(x^*)$. Both operations are executed on the fly so that $|\mathcal{Y}| = \infty$ and the learning curves are representative of both training and test performance. For the neural network, we use the same U-net architecture as in the previous experiment. We train with a batch size of 128 and a constant learning rate of $\eta = 10^{-3}$. The network updates are computed with TensorFlow's or PyTorch's automatic differentiation. Fig. 12 shows two examples from the data set, along with the corresponding inferred solutions, as well as the network learning curves for two network initializations. The measured computation time per iteration is shown in Fig. 14.

### B.5   NAVIER-STOKES EQUATIONS

Here, we give additional details on the simulation, data generation, physical gradients and network training procedure for the fluid experiment.

**Simulation details**   We simulate the fluid dynamics using a direct numerical solver. We adopt the marker-in-cell (MAC) method (Harlow & Welch, 1965; Harlow, 1972) which guarantees stable simulations even for large velocities or time increments. The velocity vectors are sampled in staggered form at the face centers of grid cells while the marker density is sampled at the cell centers. The initial velocity $v_0$ is specified at cell centers and resampled to a staggered grid for the simulation. Our simulation employs a second-order advection scheme (Selle et al., 2008) to transport both the marker and the velocity vectors. This step introduces significant amount of numerical diffusion which can clearly be seen in the final marker distributions. Hence, we do not numerically solve

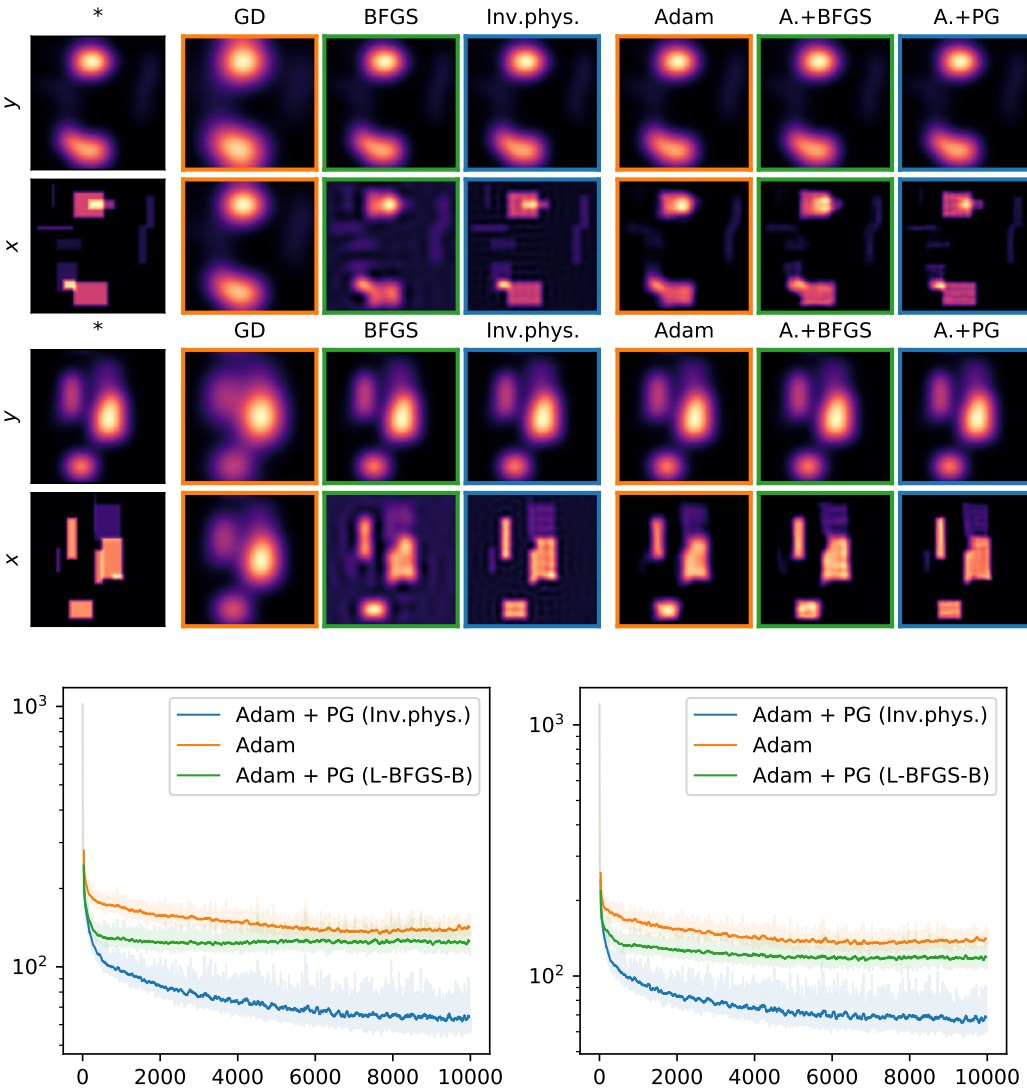

Figure 12: Inverse problems involving the heat equation. **Top:** Two examples from the data set. The top row shows observed target ($y^*$) and simulated observations resulting from inferred solutions. The bottom row shows the ground truth solution ($x^*$) and inferred solutions. From left to right: ground truth; gradient descent (GD), L-BFGS-B (BFGS) and inverse physics (Inv.phys.), running for 100 iterations each, starting with $x_0 = 0$; Networks trained for 10k iterations. **Bottom:** Neural network learning curves for two random network initializations, measured in terms of $||x - x^*||_1$.

for adding additional viscosity. Incompressibility is achieved via Helmholz decomposition of the velocity field using a conjugate gradient solve.

Neither pressure projection nor advection are energy-conserving operations. While specialized energy-conserving simulation schemes for fluids exist (Gluhovsky & Tong, 1999; Mullen et al., 2009), we instead enforce energy conservation by normalizing the velocity field at each time step to the total energy of the previous time step. Here, the energy is computed as $E = \int_{\mathbb{R}^2} dx \, v(x)^2$ since we assume constant fluid density.

**Data generation**     The data set consists of marker pairs $\{m_0, m_t\}$ which are randomly generated on-the-fly. For each example, a center position for $m_0$ is chosen on a grid of 64x64 cells. $m_0$ is then generated from discretized noise fluctuations to fill half the domain size in each dimension. The number of marked cells is random.

Next, a ground truth initial velocity $v_0$ is generated from three components. First, a uniform velocity field moves the marker towards the center of the domain to avoid boundary collisions. Second, a large vortex with random strength and direction is added. The velocity magnitude of the vortex falls off with a Gaussian function depending on the distance from the vortex center. Third, smaller-scale vortices of random strengths and sizes are added additionally perturb the flow fields. These are generated by assigning a random amplitude and phase to each frequency making up the velocity field. The range from which the amplitudes are sampled depends on the magnitude frequency.

Given $m_0$ and $v_0$, a ground truth simulation is run for $t = 2$ with $\Delta t = 0.25$. The resulting marker density is then used as the target for the optimization. This ensures that there exists a solution for each example.

**Computation of physical gradients**     To compute the physical gradients for this example, we construct an explicit formulation $\hat{v}_0 = \mathcal{P}^{-1}(m_0, m_t \,|\, x_0)$ that produces an estimate for $v_0$ given an initial guess $x_0$ by locally inverting the physics. From this information, it fits the coarse velocity, i.e. the uniform velocity and the vortex present in the data. This use of domain knowledge, i.e., enforcing the translation and rotation components of the velocity field as a prior, is what allows it to produce a much better estimate of $v_0$ than the regular gradient. More formally, it assumes that the solution lies on a manifold that is much more low-dimensional than $v_0$. On the other hand, this estimator ignores the small-scale velocity fluctuations which limits the accuracy it can achieve. However, the difficulty of fitting the full velocity field without any assumptions outweighs this limitation. Nevertheless, GD could eventually lead to better results if trained for an extremely long time.

To estimate the vortex strength, the estimator runs a reverse Navier-Stokes simulation. The reverse simulation is initialized with the marker $m_t^{\text{rev}} = m_t$ and velocity $v_t^{\text{rev}} = v^t$ from the forward simulation. The reverse simulation then computes $m^{\text{rev}}$ and $v^{\text{rev}}$ for all time steps by performing simulation steps with $\Delta t = -0.25$. Then, the update to the vortex strength is computed from the differences $m^{\text{rev}} - m$ at each time step and an estimate of the vortex location at these time steps.

**Neural network training**     We train a U-net (Ronneberger et al., 2015) similar to the previous experiments but with 5 resolution levels. The network contains a total of 49,570 trainable parameters. The network is given the observed markers $m_0$ and $m_t$, resulting in an input consisting of two feature maps. It outputs two feature maps which are interpreted as a velocity field sampled at cell centers.

The objective function is defined as $|\mathcal{F}(\mathcal{P}(x) - y^*)| \cdot w$ where $\mathcal{F}$ denotes the two-dimensional Fourier transform and $w$ is a weighting vector that factors high frequencies exponentially less than low frequencies.

We train the network using Adam with a learning rate of 0.005 and mini-batches containing 64 examples each, using PyTorch's automatic differentiation to compute the weight updates. We found that second-order optimizers like L-BFGS-B yield no significant advantage over gradient descent, and typically overshoot in terms of high-frequency motions. Example trajectories and reconstructions are shown in Fig. 13 and performance measurements are shown in Fig. 14.

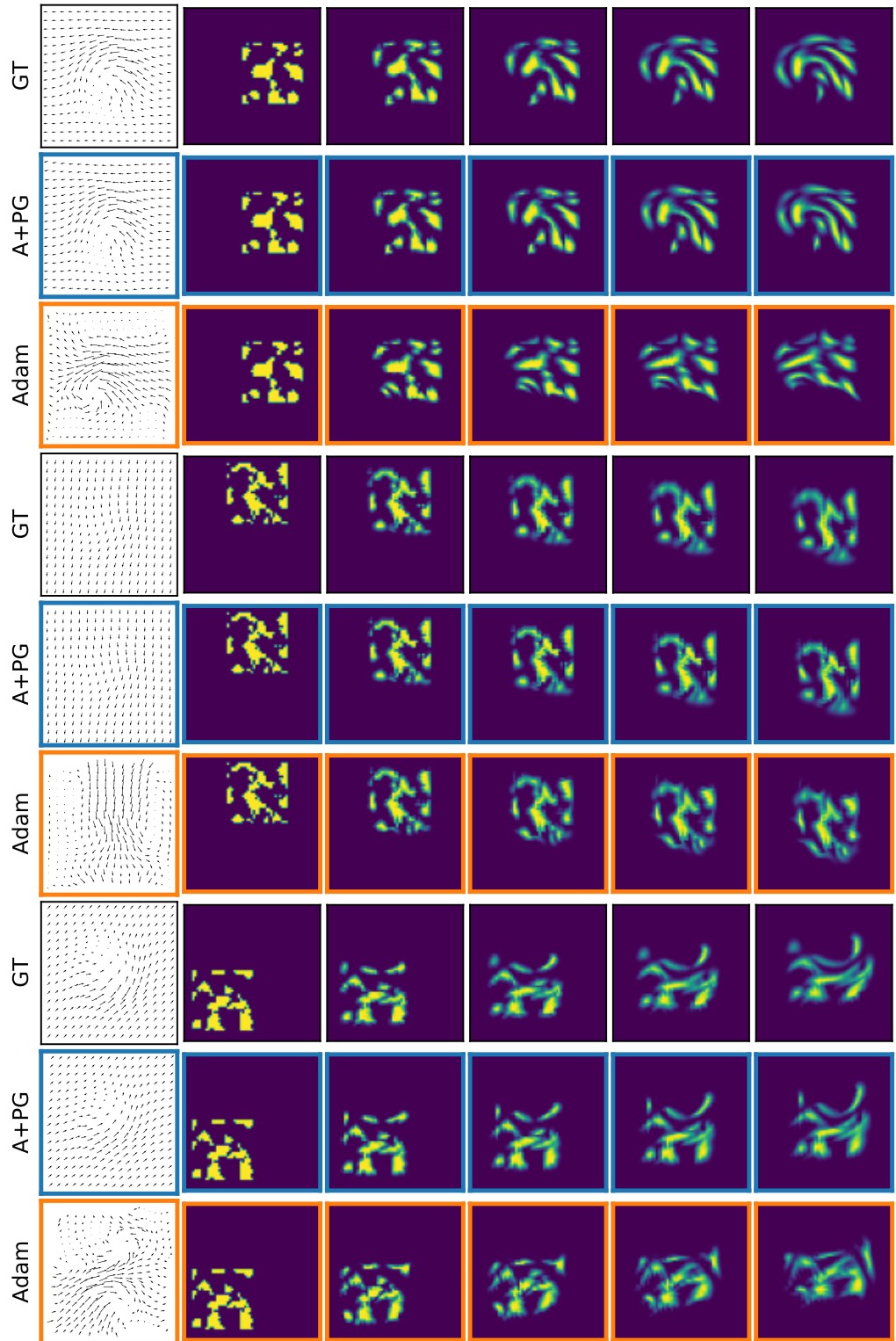

Figure 13: Three example inverse problems involving the Navier-Stokes equations. For each example, the ground truth (GT) and neural network reconstructions using Adam with physical gradient (A+PG) and pure Adam training (Adam) are displayed as rows. Each row shows the initial velocity $v_0 \equiv x$ as well as five frames from the resulting marker density sequence $m(t)$, at time steps $t \in \{0, 0.5, 1, 1.5, 2\}$. The differences of the Adam version are especially clear in terms of $v_0$.

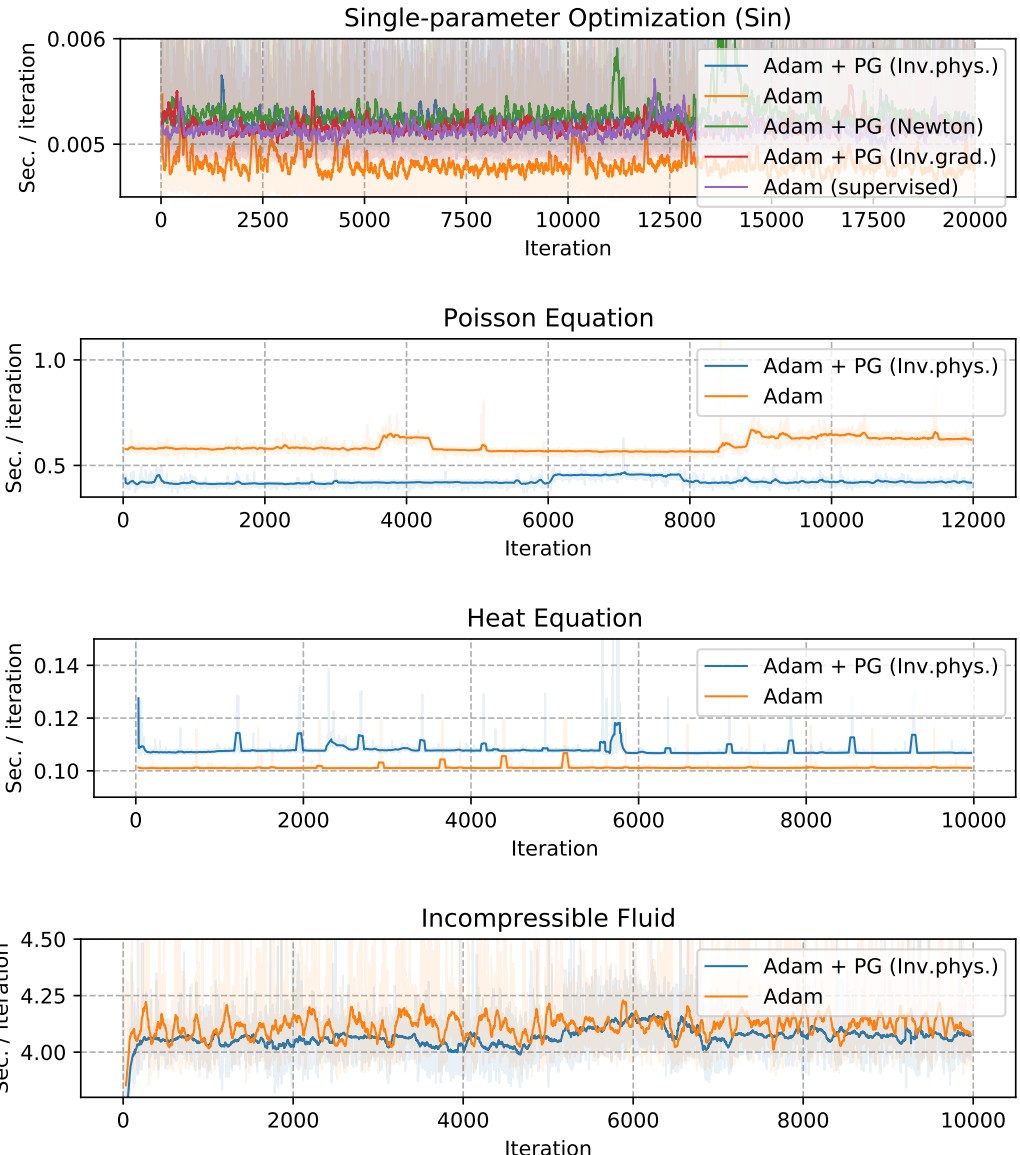

Figure 14: Measured time per neural network training iteration for all experiments, averaged over 64 mini-batches. Step times were measured using Python's `perf_counter()` function and include data generation, gradient evaluation and network update. In all experiments, the computational cost difference between the various gradients is marginal, affecting the overall training time by less than 10%.

