# OpenReview forum: "Physical Gradients for Deep Learning"
_ICLR.cc/2022/Conference — ICLR 2022 Submitted_

### Official Review · Reviewer_mue1 · 2021-10-29

**Correctness:** 2
**Technical Novelty And Significance:** 2
**Empirical Novelty And Significance:** 2
**Recommendation:** 3
**Confidence:** 2

**Main Review:**

Frankly speaking, I did not understand the main goal of this paper. I am happy to adjust my scores when the concerns are addressed and there is a high chance that I did not understand vital parts of this paper.

The main method or the goal of the paper is illustrated page 2 Section 2 with equations (1) and (2). The inverse problems are usually framed into two different frameworks:
1. Optimization framework where x* = min_x L(x). The goal is to minimize the loss function which is defined as the difference between the target y and F(x) (F is the forward physical process). This is consistent with the paper's equation (1). However, this is not what this physics gradient is meant for.
2. Mapping learning framework where we learn (usually using NN) either a probabilistic mapping or deterministic mapping from target y space to the control parameter space x. That is we learn Inv: Y -> X. This seems to be what the authors are trying to illustrate from Equation (2). However I struggle to understand the notion of "This training scheme is unsupervised, i.e. no labels are required over the training data Y". How can the mapping paradigm be learned without any training data pairs (x, y) which comes from the actual physical process? Do the authors mean the P(NN(y*|theta)) can somehow skip being evaluated by the actual physical process? If not, which means the underlying physical process is called during training, then we have the ground truth pair of (x,y) during training, which means this counts as an supervised training scheme.

Unless the authors are referring to some more advanced techniques like Physics informed neural network where the PDE equation prior is hardcoded in the loss, I really can't see why this is a unsupervised task.

Secondly, I didn't quite understand the assumption of having an inverse problem solver P^{-1} in the first place. If we have a numerical inverse problem solver, why do we need to use a neural network to solve the inverse problem anyway?  The assumption of having access to the forward numerical process P is ok, but the access to an oracle P^{-1} really made me struggle to understand.

This is the biggest concern I have, this method proposes to solve inverse problems using neural networks, but it needs access to a numerical inverse solver from domain knowledge, which seems to beats the whole purpose of using neural networks.

Finally, the empirical results. Given the above questions raised a mere misunderstanding on my part, all the "convergence speed comparison" is done on the unit of "iteration". However, I would assume that the numerical inverse solver P^{-1} would take a much longer time in wall clock sense than a neural network update. Therefore without this critical information about the actual running time of the inverse solver, the mere comparison between performance upon iterations is meaningless as the total time of Inv. Phys gradient can well exceed the Adam and even Adam + Newton.

My apology if I missed crucial parts of this paper, but I did spend some time on it and still struggle to understand things I mentioned above. If the above concerns are ultimately addressed by the rebuttal, the clarity of the delivery would need a lot of improvement.

**Summary Of The Paper:**

The paper proposed a new update method for neural network solving inverse problem by incorporating the physical information as prior into the gradient computation step. It leads to faster convergence and better convergence performance.

**Summary Of The Review:**

The basis for problem set up is unclear to me in general, therefore the contributions and results are completely compromised by that.

---

> ### Author Response · Authors · 2021-11-15
> **Response to mue1’s Review**
>
> Thank you for pointing out the unclear aspects of our paper. Below we address these points and we will, of course, update future versions of our paper accordingly.
>
> As you said, we consider inverse problems (Eq. 1) and approximating solutions to them using neural networks (Eq. 2). What we mean by “unsupervised” is that the loss formulation (Eq. 2) only contains the data y* which is given to the network. There is no x* in the equation. This is similar to autoencoders where the loss only depends on a data set of unlabelled data. Supervised learning, in contrast, requires labels for the data that is passed to the network.
> Our method is similar to supervised learning in that P^-1 projects the data into the x-space where the L2 loss for the network is formulated. The difference to supervised learning is, that with PGs, the “labels” P^-1(y*) can change during training, which we denote by the conditioning on the prediction in Eq. 5. This subtle difference has not been addressed in previous work, and can profoundly impact the learning outcomes, as demonstrated by our sine experiment.
>
> > I didn't quite understand the assumption of having an inverse problem solver
>
> While our method requires an inverse problem solver, there are several drawbacks with using only the inverse solver. First, the inverse solver must simulate the process at least once, probably many times to find a solution. This is typically far more expensive than the optimization algorithm itself. A neural network learns to approximate the solution without performing the simulation and the solution inference is, thus, much faster.
> Second, we don’t require the available inverse solver to initially converge for all examples. As long as its updates don’t explode within one iteration, we can still use it to train the network.
>
> > All convergence speed comparison is done on the unit of "iteration"
>
> In all our examples, the time cost per iteration is roughly equal for all methods under consideration. More detailed performance graphs are shown on page 26. We use iterations as the base for our comparisons because we want to highlight the differences due to the update steps performed by each method. The improvement per update step is much greater with PGs than, e.g., Adam because the search direction is fundamentally improved. Plotting against time could add secondary effects that alter the diagrams, making the interpretation more difficult. However, in our experiments the loss plots against time look extremely similar. Nonetheless, we will add them to the appendix in our next revision.
>
> >  I would assume that the numerical inverse solver P^{-1} would take a much longer time
>
> The computation time depends on the problem. For most of our experiments, there was no considerable difference, with a couple of exceptions: 1.) The 32-step BFGS solver in the heat experiment takes much longer to compute an update than Adam and inverse physics training. 2.) The inversion in the Poisson experiment is faster than computing the regular gradient required by Adam.
> Inversion with domain knowledge is typically roughly as expensive as the simulation itself, possibly cheaper if some properties of the governing equations can be exploited. When compressing multiple update steps into one (n>1), the cost rises linearly so this only pays off for cheap simulations.
>
> If you have further questions, please feel free to post them here, and we are happy to answer and clarify as soon as possible.

---

> > ### Comment · Reviewer_mue1 · 2021-11-19
> > **more discussion needed**
> >
> > First of all, I don't think I would call that a "unsupervised" training merely due to the lack of labels in x space. To me, "unsupervised" does not mean you don't have an explicit MSE term on the end goal, but comparing to "supervised' you do not need labels that require extra effort from simulator or humans. VAE is unsupervised as there are no extra human labels needed. To me, this is very much supervised by the presence of forward simulator in the loop, although the loss is not a MSE term on two x-space values, it used the supervision of f(x) where f() is the simulator.
> >
> > Secondly, I agree that PG + NN is better than the inverse solver itself, but it fails to show me that this is better than any other existing deep inverse solvers like Neural-adjoint method.
> >
> > Thirdly, Why is the speed of Adam + PG faster than Adam along in the Poisson equation case in page 27? If the analytical inverse problem solver can be done faster than a neural network backward pass, would that mean the inverse problem, would that mean the inverse problem is so unique that it does not need a neural network in the first place?
> >
> > Thank you for your response, really appreciate the discussion and help for me to understand the paper

---

> > > ### Author Response · Authors · 2021-11-19
> > > **Response to mue1's comment**
> > >
> > > > I agree that PG + NN is better than the inverse solver itself, but it fails to show me that this is better than any other existing deep inverse solvers like Neural-adjoint method.
> > >
> > > The neural-adjoint is analogous to Adam training in our setting and we compare our results to Adam training in all experiments. In our setting, the forward simulator is known, so there is no need to use neural-adjoint.
> > >
> > > > Why is the speed of Adam + PG faster than Adam along in the Poisson equation case in page 27?
> > >
> > > Note that these are times measured for training, not inference. The Poisson problem is indeed special. Its inverse can be written as a sparse matrix. The backwards pass for the domain specific inversion only requires a matrix multiplication, making the computation very efficient. The forwards and backwards pass for the differentiable solver with Adam require matrix inversions, which is why Adam + PG is faster by using only the matrix multiplication in this case.

---

### Official Review · Reviewer_ML7L · 2021-11-02

**Correctness:** 2
**Technical Novelty And Significance:** 2
**Empirical Novelty And Significance:** 2
**Recommendation:** 3
**Confidence:** 4

**Main Review:**

The omnipresence of inverse problems asks for stable and efficient deep learning-based solvers and new methods are highly desired.
While the proposed usage of so-called physical gradients is based on a rather minor modification of the supervised approach, it could overcome several drawbacks of existing methods and thus seems to be a research direction well worth pursuing.
Unfortunately, the present paper does not sufficiently motivate and justify the usage of physical gradients and has several other issues.


- **Motivation:**

    The proposed method employs an iterative solver in each optimization step in order to create the current targets for the neural network. Thus, it seems that the neural network cannot outperform the iterative solver except for perhaps averaging out noise or adapting the initialization. This raises the question, why one should prefer the neural network over the iterative solver?  Similar to the supervised approach, it might be beneficial to replace the iterative solver by a neural network, as this could lead to faster inference times in case of a computationally expensive solver and allow for sensitivity analysis / uncertainty estimation via the trained neural network. The former, however, would drastically slow down the neural network training when using physical gradients as the iterative solver needs to be applied in every optimization step and the latter is not explored in the paper.


- **Empirical evidence:**

    Unfortunately, the numerical experiments do not showcase the potential strengths of the proposed method. While a crucial part of this method is the initialization of the iterative solver using the current network prediction, this seems to only be used in the first toy example. In the other examples, the difference between the use of physical gradients and the supervised approach is not clear. Moreover, the proposed method might also work with only a few iterations of the iterative solver in each step. However, for all but the last example, an explicit solution is used instead of an iterative solver and there is no numerical evidence of how the method depends on the accuracy of the iterative solver.  Finally, all experiments use an infinite stream of synthetically created (mostly noiseless) data, which is usually not given in practice.


- **Further comments:**

    Further issues, comments, and suggestions for improvements can be found in the following:

    (i) The statement that physical gradients behave more smoothly, both in magnitude and direction, appears in Subsection 3.4 and in the abstract. However, apart from the toy example in Figure 7, it seems that there is not enough theoretical or empirical evidence for that statement.

    (ii) In Equations (1) and (2), $x^*$ and $\theta_*$ belong to the $\operatorname{argmin}$ (which is in general a set of minimizers, given that the minimum is attained), rather than being equal to the $\min$.

    (iii) I guess that $\mathcal{P}^{-1}_{n^*}(y^* | x_0)$ should, in general, not be equal to $x^*$ as stated in the introduction of Subsection 2.1. It should probably only be an approximation of the latter.

    (iv) In my opinion the sentences "Note that the dimensions of ..." and the subsequent one, are rather confusing. What is meant by "dimensions" in this context, why is this phenomenon counter-intuitive, and how does this relate to the vanishing/exploding gradient problem in *deep* networks?

    (v) It seems that the derivative of the forward operator is missing in the update rules in Figure 1 and also the expression for the Hessian given in the caption should be derived in more detail.

    (vi) I am not aware of a generic definition of a "sensitive" function (as mentioned several times in the paper) and think that this should be properly defined.

    (vii) The statement that jointly optimizing all inverse problems reduces the likelihood that an individual solution gets stuck in a local minimum, should, if it holds true in this generality, be referenced or sufficiently motivated.

    (viii) In Algorithm 1, it is unclear what it means to condition the inverse solver on two values $x_{k-1}$ and $y_k$.

    (ix) The first example does not fit into the framework described in Equations (1) and (2) as now $\mathcal{P}$ is parametrized by $a$ and $y^*$ is fixed.

    (x) In the last example, $y^*$ seems to correspond to the tuple $(m_0,m_t)$ rather than only $m_t$.

    (xi) It is not clear, how Assumption 3 on page 15 relates to the universal approximation theorem (as stated below), which deals with uniform approximation of a continuous function on a compactum rather than convergence of gradient descent.

    (xii) The factor $1/2$ seems to be missing in the loss $L$ in Subsection B.2.

    (xiii) In Subsection B.4 there appears $x_0$ which seems not to be defined.

    (xiv) Typos: *an* generic inverse solver; *we* found that second-order optimizers

**Summary Of The Paper:**

The present paper proposes a novel ansatz for solving inverse problems by means of deep learning. Specifically, the loss attached to a specific observation is defined as the difference between the neural network prediction and the output of an approximate iterative solver initialized with the current neural network prediction. The gradient of such loss (for a mini-batch of observations) w.r.t. to the neural network parameters, coined physical gradient, can then be used to optimize the neural network by standard variants of gradient descent. Potential advantages of this approach over the classical (un-)supervised approaches include the following: (1) One circumvents the need for (possibly ill-conditioned) automatic differentiation of the forward operator, (2) the varying initialization of the iterative solver enables adaptive convergence to different solutions (in case of a non-injective forward operator) throughout the neural network training, and (3) a few iterations of the solver in each step might suffice to provide a useful loss for the neural network. Moreover, the paper also explores the use of (Quasi-)Newton methods.
The functionality of the proposed method is backed by theoretical results and four numerical experiments.

**Summary Of The Review:**

The paper proposes an interesting adaptation to the supervised approach of solving inverse problems by means of deep learning. Unfortunately, the included experiments are not representative enough to show whether the promised benefits of this approach hold true. In addition, there are several issues regarding the presentation of the material.

---

> ### Author Response · Authors · 2021-11-15
> **Response to ML7L’s Review**
>
> > the neural network cannot outperform the iterative solver
>
> It is reasonable to assume that an iterative solver can converge to any desired accuracy assuming the initial guess is close enough to an optimum. Thus, our method can also reach any accuracy. However, iterative solvers often struggle when the initial guess is further away from an optimum and there, our method performs better. In our general comment above, we show this on a wave packet fitting task. There, the iterative solver fails to locate the wave packet in a considerable number of cases and sometimes diverges completely while the network trained on the same objective learns to fit all wave packets.
>
> > the difference between the use of physical gradients and the supervised approach is not clear
>
> As we explain in our general comment, the difference between physical gradients and supervised learning is that the “labels” change during PG training. This is very important because it eliminates the arguably biggest weakness of supervised learning, its inability to deal with multimodal data.
> In the heat and Poisson experiments, there is a unique ground-truth solution. This solution is not used by any of the methods but it allows us to unambiguously evaluate the quality of the inferred solutions. More complex systems are often multi-modal and supervised training performs poorly in these conditions as we show in our sine, fluid and the wave packet experiments.
>
> > all experiments use an infinite stream of synthetically created (mostly noiseless) data
>
> The infinite stream of i.i.d. data makes the interpretation of the results very clear because there is no overfitting. We are interested in the fundamental differences between the various optimizers and believe that this can best be demonstrated when there are no other factors hampering the learning. Since a simulation of the process is a requirement for our method, it is generally possible to generate arbitrary amounts of training data.
> The learning curves we show represent the performance usually evaluated on a test data set but without the bias that typically comes with a finite test set.
>
> > The first example does not fit into the framework
>
> That is an oversight on our part. The equations were a bit different in earlier versions. We will repeat the experiment with y varying instead of a. This will only have a small impact on the relative performance of the different methods.
>
> > It is not clear, how Assumption 3 on page 15 relates to the universal approximation theorem
>
> The universal approximation theorem states that a sufficiently large neural network can approximate any continuous function. With a finite number of distinct data points, such a neural network can consequently map each data point to an arbitrary output. This is what we require for assumption 3: the network, f, needs to be able to converge to an arbitrary label x* for each input y under gradient descent update steps.
>
> Thank you for the detailed suggestions under “further comments”. Our next revision will also include updates addressing these comments.

---

> > ### Comment · Reviewer_ML7L · 2021-11-21
> > **Response to authors' clarification**
> >
> > Thank you for your response and the updated manuscript. I highly appreciate your effort in including all of the reviewers' feedback, which also alleviated some of my concerns.
> >
> > #### **NEW EXAMPLE**
> > The new example shows the advantage of solving an *unsupervised* task with squared loss using a NN on a set of observations in comparison with using L-BFGS-B on a single observation.
> >
> > This seems, however, not to answer my question
> >
> >     the neural network cannot outperform the iterative solver
> >
> > with respect to the *supervised and PG* approach.
> >
> > The main concern has been that NNs trained on *labels produced by the inverse solver*, which is done in supervised as well as in PG approaches, might not been able outperform the inverse solver.
> >
> > #### **MISSING EVIDENCE**
> > I would strongly suggest to showcase the *performance of PG* with examples which use an *iterative* inverse solver *conditioned on the NN predictions*.
> > The performance of PG should be evaluated in terms of accuracy metrics and training/inference time (mean & standard deviation over multiple runs) against
> >
> > - the iterative solver itself (averaged over the same samples)
> > - the unsupervised method
> > - the supervised method (using the same iterative solver)
> > - if available, another domain-specific neural network approach from the literature.
> >
> > For the iterative solver, the supervised method, and PGs, the effect of the number of iterations should also be analyzed.
> >
> > None of the presented examples seem to cover all of this in a realistic setting.
> >
> > Therefore, it is still hard for me to judge the benefit of using PGs and hence the contributions of the present paper.
> >
> > #### **GROUND-TRUTH SOLUTION**
> > If I understand it correctly, the unique ground-truth solution of the heat and Possion experiments is used in the expressions for the PGs. Could you please elaborate on your comment that
> >
> >     This solution is not used by any of the methods [...].
> >
> > #### **ASSUMPTION 3**
> > I agree that a sufficiently large network can fit arbitrary data, however, it is not clear that such a global optimum can be found via Gradient descent. Of course, there exists corresponding literature in the overparametrized regime, which, however, needs further assumptions.
> >
> > #### **OVERALL CLARITY**
> > As also mentioned by other reviewers and demonstrated by the discussions, I think that the material needs to be presented in a clearer and more accessible fashion.

---

> > > ### Author Response · Authors · 2021-11-21
> > > **Response to ML7L's comment**
> > >
> > > Thank you for your additional feedback and suggestions! We appreciate the additional effort you put into your review.
> > >
> > > > the neural network cannot outperform the iterative solver
> > >
> > > Given unlimited time and a good initial guess, iterative solvers will eventually find the best-fitting solution in terms of loss value. While the smoothing done by neural networks can, in some cases, improve accuracy in solution space beyond what an iterative solver can achieve, this phenomenon lies outside the scope of our current work. We assume that the iterative solver is capable of finding the correct solution, which is the case in all of our experiments. We focus on the advantages in terms of speed and stability, and here NNs can definitely outperform iterative solvers.
> > >
> > > > If I understand it correctly, the unique ground-truth solution of the heat and Possion experiments is used in the expressions for the PGs
> > >
> > > No, the ground truth solution is not available to any of the methods and only used in the evaluation of the results. The PG formulations in the heat and Poisson example try to recover something that gets as close as possible to the ground truth solution. In the Poisson case, that can be easily done analytically to high precision but in the heat experiment, the inversion is unstable and requires special handling. We describe the specifics of how we invert the heat equation in B.4 “Stable physical gradients” but the result is only a rough approximation of the ground truth solution.
> > >
> > > > I agree that a sufficiently large network can fit arbitrary data, however, it is not clear that such a global optimum can be found via Gradient descent. Of course, there exists corresponding literature in the overparametrized regime, which, however, needs further assumptions.
> > >
> > > Good point. While the formulation of assumption 3 itself is correct, we will clarify the explanation below, citing works on overparametrized fitting / learning.

---

> > > > ### Comment · Reviewer_ML7L · 2021-11-22
> > > > **Reponse to Authors' comment**
> > > >
> > > > Thank you very much for the explanations.
> > > >
> > > > #### **PG vs. ITERATIVE SOLVER vs. SUPERVISED APPROACH**
> > > >
> > > > I think that the manuscript should emphasize
> > > > 1. the assumption of a perfect inverse solver (given a suitable initialization)
> > > > 2. the main goal of enhancing *inference time* and stability.
> > > >
> > > > Moreover, it should also be noted that the *training time* of PG is much larger than the training time of the supervised approach, due to the fact that one needs to recalculate every label in each iteration.
> > > >
> > > > I wonder why PG is compared to the supervised approach in only one example. This makes it difficult to judge whether PG is consistently more stable.
> > > >
> > > > #### **GROUND-TRUTH SOLUTION**
> > > >
> > > > I guess it is more a matter of definition whether the PG *uses* the ground-truth solution. In order to
> > > >
> > > >     [...] recover something that gets as close as possible to the ground truth solution
> > > >
> > > > an expression of the ground-truth solution at least seems to appear in the PG, e.g., $\mathcal{F}^{-1}\big(\mathcal{F}(y)e^{k^2(t_*-t_0)}\big)$ and $\nabla^2 y$.

---

> > > > > ### Author Response · Authors · 2021-11-22
> > > > > **Response to ML7L's comment**
> > > > >
> > > > > Thank you for your quick response!
> > > > >
> > > > > > I think that the manuscript should emphasize 1. the assumption of a perfect inverse solver [...] 2. the main goal of enhancing inference time and stability.
> > > > >
> > > > > Yes, our previous changes to the manuscript already work towards that goal. We will continue to work on the manuscript and more clearly explain the assumption of a perfect inverse solver (currently stated in Assumption 2 in A.2) in the main text as well.
> > > > >
> > > > > > it should also be noted that the training time of PG is much larger than the training time of the supervised approach
> > > > >
> > > > > The time *per iteration* is larger with PGs. We will clarify that to the results section and add a short discussion to the conclusion section.
> > > > >
> > > > > > I guess it is more a matter of definition whether the PG uses the ground-truth solution
> > > > >
> > > > > Right, if the ground truth solution can be recovered from the observations, you may say that it is being used. However, we never explicitly pass that data to the method. This is to account for settings where no ground truth solution is available but only recorded data.

---

### Official Review · Reviewer_FRkr · 2021-11-04

**Correctness:** 2
**Technical Novelty And Significance:** 3
**Empirical Novelty And Significance:** 4
**Recommendation:** 6
**Confidence:** 4

**Main Review:**

Strengths:
* The idea the paper proposes is simple but powerful: That writing down a particular (intuitive) proxy L2 loss for the original inverse problem yields stable gradient updates with direct physical intuition in the physical output space.
* The empirical demonstrations are extensive, and demonstrate the superior performance of the proposed method in physical settings with different salient properties (e.g., multi-modality, large sensitivity, singularities, etc.)

Weaknesses:
* The paper is written so as to significantly oversell the generality of the proposed framework, which in turn significantly impacts clarity. There are several major examples of this:
	* The title "physical gradients" and several phrases throughout (e.g. in the abstract: "We replace the gradient of the physical process by a new construct") strongly imply that a new method of taking gradients will be introduced. This is not the case. Instead, the original inverse problem is rewitten such at the gradients that "fall out" have physical intuition.
    * The formulations in Equations 4 and 5 depend on $\mathcal{P}_n^{-1}$, which in various places is described as being derived using domain knowledge. However, $\mathcal{P}_n^{-1}$ is just a single step of gradient descent, and while this is mentioned, it is not emphasized. Notably, all experiments use this "single step of gradient descent" interpretation, and the physical interpretation of the resultant gradient depends on this as well. This point should be made much clearer.
    * Relatedly, the abstract indicates that the training approach "combines higher-order optimization methods with machine learning techniques." To the best of my knowledge, this does not seem to be true.
* Certain claims made in the paper are not fully substantiated. For instance:
    * The claims in Figure 1(b) about the strengths of the present method vs. GD and Newton's method are not fully substantiated/described.
    * On page 4, it is noted that the method can be applied under one of three conditions. However, only condition (iii) is ever explicitly used (as far as I understand), and the rationale as to why either of the other two conditions is sufficient is never described.
* The main body of the paper is not self-contained, causing issues with clarity that are then exacerbated by typos. In particular, Appendix A.1 should definitely be in the main body of the paper in order to make the overall method/intuition understandable. (Also, the last paragraph of page 4 mentions that x_n can be taken as a constant - this is, importantly, not the case, but requires digging into Appendix A.1 in order to understand.)

Minor comments (not affecting my score):
* The paragraph on "inverse physics via domain knowledge" is hard to understand until after the entire paper has been read; its clarity could be improved.
* Above equation 3: The authors might consider using a different subscript than "sup." Presumably this is meant to stand for "supervised," but could be confused for "supremum" given the optimization setting.
* In the experiments, the descriptions of what methods are being compared could be made much crisper and clearer. In addition, interpretation of the legend in e.g. Figure 2 should be made more self-contained.
* In all graphs, legends should be moved such that the lines are visible.
* In Figure 3, it is not clear what the x-axis is. The caption of 3(a) presumably has a typo, as convergence curves are not shown.

**Summary Of The Paper:**

This paper proposes an approach for learning neural network-based approximators for inverse problems in physical domains. The crux of this approach involves rewriting the training loss function to capture the difference between (a) the NN's estimate of the problem input and (b) the solution obtained when using this estimate as an initial point for an approximate inverse problem solver. This loss function is then optimized via gradient descent. This approach is demonstrated on several physical systems, and shown to avoid issues due to multi-modality (which supervised training schemes fall prey to), instability in the gradients through the physical process (faced in some settings by first-order training methods applied to the "standard" inverse problem), and issues such as expense of computation or ill-conditioning (faced in some settings by training method using e.g. inverse Hessian information).

**Summary Of The Review:**

The method provided by this paper is simple, powerful, and physically intuitive. However, there are significant issues with overselling, substantiation of claims, and clarity that mean I cannot recommend acceptance of this paper in its current form.

---

> ### Author Response · Authors · 2021-11-15
> **Response to FRkr’s Review**
>
> You are correct in stating that we do not derive a new method for computing gradients. Instead, our method uses existing optimizers and re-interprets their updates as gradients for a gradient-descent-based optimization, i.e. for network training. We call them physical gradients because they replace the traditional gradient that would be used for the physical process and they are derived from a physically-based optimization. We will re-formulate our paper based on the feedback to clarify this aspect of our method.
>
> > $P^{-1}$ [...]  is just a single step of gradient descent
>
> There might be a misunderstanding here caused by the notation. We use the subscript to distinguish between different versions of inverse solvers. $P_n$ means performing n iterations and n* is the minimum number of iterations to fully converge. We discuss the case of using a single gradient-descent step to show how our method relates to traditional differentiable physics. Our next revision will make this distinction clearer.
> In our fluids experiment, we use a domain-knowledge-based solver with n=1, in the sine experiment, we show Newton’s method with n=1, and in the heat experiment, we show BFGS with n=32.
>
> > the abstract indicates that the training approach "combines higher-order optimization methods with machine learning techniques."
>
> We do this in experiment 1 (sine) with Newton’s method and in experiment 3 (heat) with L-BFGS-B. The domain-knowledge-derived updates represent higher-order or potentially infinite-order update steps, leading to the very significant accuracy improvements shown in our paper.
>
> > The claims in Figure 1(b) [...] are not fully substantiated/described.
>
> Assuming you are referring to the check marks for sensitivity, hessian-dependence and non-linearity, this classification describes properties of the update steps themselves. We agree that the discussion in the main text was not sufficient and we will expand it in the next revision.
> The PG properties depend on what specific physics optimizer is used which was not properly communicated in the table. However, simply using two gradient descent steps for the PG checks all the boxes, so we feel the check marks are justified. We will annotate the check marks to make this clearer.
>
> > On page 4, it is noted that the method can be applied under one of three conditions.
>
> You are right in that this claim warrants further discussion. We focus on condition (iii) in our experiments. We use (ii) in the sine experiment to learn with Newton updates using an analytic formulation and a variant of (i) in the heat experiment when training with BFGS. If (i) or (ii) are fulfilled, one can compute a Newton update, or chain multiple updates, and use them as the PG for network training.
>
> > The main body of the paper is not self-contained
>
> You are right in that Appendix A.1 contains some important observations. We will integrate them into the method section but will leave the proof in the appendix.
>
> Our next revision will also contain corrections in accordance with your minor comments.

---

> > ### Comment · Reviewer_FRkr · 2021-11-20
> > **Good response**
> >
> > Thanks to the authors for their clarifications and revisions to the paper, which are indeed helpful.
> >
> > Some residual questions:
> > * Section 3.1 and 3.2: What is the benefit of the PG method if an analytical inverse solver exists? I understand that these are toy examples (and e.g., non-invertible physics are demonstrated in the later experiments), but it doesn't make sense to me to use an analytic solver in the loop of training if the analytic solver itself could just be used in the first place.
> >
> > Additional clarity suggestions:
> > * The "physical gradient" should be more precisely defined, in order to improve overall clarity. The definition is slightly different in different places (e.g. page 2, "The result of the embedded inverse-problem solver takes the place of the first-order gradient of the physical process, and we refer to it as the physical gradient (PG)." vs. page 5, "the difference between prediction and correction obtained from $\mathcal{P}^{-1}_n$, which we refer to as the physical gradient")
> >
> > Based on the revisions and responses, I have raised my review. However, since my other chief concern besides overselling was clarity, it is difficult to further raise my review without seeing the clarity revisions. In particular, while I now understand the contributions given both the manuscript and the content of the review discussion, I think the paper itself does not yet "stand alone" in terms of clarity.

---

> > > ### Author Response · Authors · 2021-11-21
> > > **Response to FRkr's comment**
> > >
> > > Thank you for having another detailed look at our paper! We appreciate the effort you have put into your review.
> > >
> > > > What is the benefit of the PG method if an analytical inverse solver exists? I understand that these are toy examples [...]
> > >
> > > Correct, the heat and Poisson experiments can be efficiently solved with inverse solvers as well. However, they highlight specific properties that also occur in more complex problems where no such inverse solver exists. Neural networks scale to these more complex problems (like our fluid example) while inverse solvers do not. Studying these simpler examples first therefore is an important and natural step to understand the behavior in more complex situations.
> > >
> > > > The "physical gradient" should be more precisely defined
> > >
> > > The definition on page 5 is correct. We will reformulate other parts of our manuscript to make the definition clearer.
> > >
> > > We will also keep working on the overall clarity of the paper. If you have other points or specific passages in mind that were hard to understand, please let us know.

---

### Official Review · Reviewer_pjrC · 2021-11-05

**Correctness:** 4
**Technical Novelty And Significance:** 3
**Empirical Novelty And Significance:** 3
**Recommendation:** 6
**Confidence:** 3

**Main Review:**

This work is clearly presented and proposes an interesting idea: using inverse solvers to provide targets that iteratively improve the neural network predictions of the initial state.

The case n=1 used is probably the most interesting, because the inverse step is indeed simply providing a slightly improved target to supervise the training of the network, without the need of labeled data and without having to run a full inverse solve.

The experiments performed, though on fairly simple domains, demonstrate that the method works and is able to generalize at least in the given domains (ie interpolate), given that data is sampled randomly.

As with many of these types of applications of machine learning to physical inference, it is questionable if the benefit of the fully trained model achieved at the end compensates the cost of training it for a long time on a limited domain, while also having to provide an already existing inverse solver (or something capable of approximating this inverse). It would be good for the authors could discuss further what use cases they foresee for this type of model in practice, where the costs of training and formulating a known inverse could outweight simply using a traditional methods. For the cases where the inverse is actually known, it would be important to provide a comparison to simply providing the inverse optimization.

-------

After reading the authors response, I maintain my previous evaluation.

**Summary Of The Paper:**

This paper looks at the (inverse) problem of finding the some initial state given a final state of a physical system. It proposes training neural networks to predict the initial state for a given final state. These networks are trained iteratively by using inverse solvers (based on domain knowledge) that provide physical "targets" that are closer to the desired initial state than the current prediction.

**Summary Of The Review:**

Despite some of the comments above, I believe the conceptual contribution and the experiments presented in this paper are of enough interest and robustness to warrant its acceptance. I am rating it as marginally above the acceptance threshold for now, expecting the authors to improve and respond to the comments above.

---

> ### Author Response · Authors · 2021-11-15
> **Response to pjrC’s Review**
>
> As you noted, we didn’t really explain the motivation for our work: Why to use neural networks in the first place? When there is a unique solution and it can quickly be computed exactly, there is of course no need to train a network. In general, however, there are two main reasons to use neural networks, which we discuss in more detail in our general comment above. First, neural networks are generally more stable than iterative optimizers because they are jointly trained on all data. Iterative solvers typically require initial guesses that are already close to a solution. Second, neural network inference is much faster than running an iterative solver for each example.
>
> We chose the Poisson and Heat experiments to illustrate various difficulties real-world problems pose, while at the same time being simple enough to intuitively understand. Despite their simple appearance, the involved challenges, like numerical instability in the inversion, make these problems highly non-trivial regarding the aspects we are highlighting.
>
> In more realistic settings - like with our fluid and wave packet example - traditional solvers take longer (>10,000x in our fluids example) and are often less reliable than a trained network.

---

### Author Response · Authors · 2021-11-15
**Clarifications Concerning the Motivation of our Work**

A big thank you to all the reviewers! You clearly put a lot of time and effort into your reviews and we appreciate the constructive, high-quality feedback.

The initial draft of the paper was clearly missing some important evaluations concerning the motivation of our work and we would like to clarify the resulting misunderstandings.

Our method has two fundamental advantages over classical iterative solvers: Stability and Speed.

1. Stability. Classical solvers can easily get stuck in local minima or completely fail to converge. Neural networks trained on the same objective but jointly on all data can successfully avoid these outcomes. Unfortunately the initial draft of our paper did not sufficiently discuss this.
In our fluid experiment, for example, the iterative gradient descent solver always gets stuck in non-optimal solutions due to the highly complex loss landscape. Even after running it for hours, it is more likely to diverge than find the correct solution. The networks always find a solution, even though it may not be optimal in the case of Adam.

In response to your comments, we have set up a new test case to show that networks given the same objective as iterative solvers converge much more reliably because they jointly optimize all available data.
The task is to localize a wave packet $A \cdot \sin(f \cdot x) \cdot exp(-\frac 1 2 (x-x_0)^2 / \sigma^2)$ and determine its amplitude from a noisy recorded time series. This kind of task is probably the most ubiquitous in physics. Picture a detector measuring laser / magnetic / seismic / gravitational pulses or quantum mechanical particles.
On this task, iterative solvers fail to reconstruct the wave packet when the initial guess is not already close to the true parameters, even diverging in 0.12% of our cases. In contrast, a neural network trained on the same data using the same objective quickly learns to fit all wave packets. We will add this experiment to our introduction to provide additional motivation for our work.

2. Speed. This is important for time-critical applications (like deciding whether to store or discard event recordings at the LHC) and applications in general that deal with large amounts of data (like gravitational wave detectors). These experiments already use neural networks for performance reasons. Unfortunately our paper didn’t provide any reference for the speedup in our experiments. We have now measured the speedup in our fluid experiment and will add the following table in the next revision.

Time to reach equal solution quality (measured as the MAE in x-space):

| Method                     | Training time       | Inference time per example |
|----------------------------|---------------------|----------------------------|
| Network                    | 17.6 h (15.6k iter) | 0.11 ms (immediate)        |
| Domain-knowledge solver    | n/a                 | 2.2 s (7 iter)             |
| Gradient descent optimizer | n/a                 | > 4h (20k iter)            |

To reach the same solution quality as the network prediction, our domain knowledge solver needs more than 10,000 times as long because it needs to run the forward and backward simulation a couple of times. With both iterative solver and network, we used a batch size of 64 and divided the total time by the batch size.

We discuss differences between our method and supervised learning in the next comment.

---

> ### Author Response · Authors · 2021-11-15
> **Differences to Supervised Learning**
>
> There also seemed to be some doubts as to how different our method is from supervised learning. As many previous works have shown, supervised learning is very effective for learning classification and regression in many applications and network architectures have been tuned to work well with it. One of its main drawbacks is that it cannot properly deal with multimodal data. In the realm of physics and engineering, this is a major limitation.
> Differentiable simulations have become very popular in these fields due to the fact that they do not suffer from this drawback. Their updates, being based on gradient descent, suffer from various other problems, however, as we describe in our paper. With increasing complexity and non-linearity of the physical process, learning with differentiable physics and gradient descent becomes increasingly difficult.
>
> Our solution to this is using differentiable simulations but with better gradients. The updates to the neural networks should resemble supervised updates for unimodal problems but not suffer from the same problems in multimodal settings.
>
> This is exactly what physical gradients do. The inversion of the physics combined with the $L_2$ loss in x-space will recover supervised updates for unimodal problems if the inversion is perfect. In multimodal settings, the conditioning of the inversion on the network prediction results in “changing labels” that guide the network to a solution.
> The difference to supervised learning is illustrated in our sine experiment where supervised learning is worse by a factor of around 1000.
>
> We will upload a revised version of our manuscript in the next couple of days. For the more detailed comments, we respond to the individual reviews below.
> Please feel free to ask further questions and we will answer them as soon as possible.
>
> Edit: We have now posted a revision of our manuscript which addresses most of the issues that were pointed out. We will keep working on the paper and post another revision before the deadline.

---

> ### Comment · Reviewer_mue1 · 2021-11-19
> **Following up on the time comparison**
>
> First of all, thank the authors for their hard work in the past week, the graphs are realy well made and this is a long appendix (which means a lot of effort)
>
> I still have questions regarding this table of speed. If you are in a application that requires speed during test time, why not compare to a pure neural network? There are a lot of deep inverse models that are based on neural network that have nearly instant inference time, like Tandem model (which by the way is very similar to some of this), cVAE, flow-based models, Neural-adjoint etc.. See "Benchmarking deep inverse models over time, and the neural-adjoint method" by Ren et al. in NeurIPS 2020 for more details.
>
> I think one important misunderstanding as a reader is which method is this PG actually comparing with? It is a neural network solving inverse problem but it compares with the traditional solver for speed, rather than the other neural networks (in either speed or performance).

---

> > ### Author Response · Authors · 2021-11-19
> > **Response to mue1's comment**
> >
> > > If you are in a application that requires speed during test time, why not compare to a pure neural network?
> >
> > In all of our experiments, we compare to neural networks trained using state-of-the-art optimizers. So there will be no difference in speed at inference time. The difference lies in the accuracy of the inferred solutions.
> >
> > > There are a lot of deep inverse models that are based on neural network. [...]  See [...] Ren et al. in NeurIPS 2020
> >
> > The methods you are referring to do not have access to the forward function but must learn from data alone. While the task discussed there is also very interesting, it poses completely different challenges than our setting.
> >
> > > which method is this PG actually comparing with?
> >
> > Our method competes with standard neural network training. Therefore we compare our method to state-of-the-art neural network optimization. We also added some comparisons to iterative solvers to highlight the benefits of using neural networks in the first place.

---

### Comment · Reviewer_mue1 · 2021-11-19
**More discussion needed...**

Dear other reviewers and author,

Thank authors for detailed response, I think one concern some of our reviewers have is around the basic understanding (significance, novelty and usefulness?) of this method.

After the authors response, I think I still failed to understand the motivation. Using neural networks, a lot of deep inverse models tries to learn directly the mapping from y to x space, see this below paper where it benchmarked a wide range of deep inverse solutions, this sounds like the "Tandem" model with a actual physical model replacing the forward proxy.

Ren et al. "Benchmarking deep inverse models over time, and the neural-adjoint method", NeurIPS 2020

Also the numerical evidence is confusing to me:
1. Why did the Newton method and Adam converge to drastically different performance? I think if the optimizer is well tuned, adam should converge to as good as a result as the Newton method although using more iterations, since the loss surface is the same? (This suggested to me that the optimizers are not well tuned for comparison)

2. Wall clock time of the experients: Second order information is definitely useful in optimization and they converge in fewer iterations than first order ones, however, it is not the mainstream optimizers for a single reason, computational speed. The wall clock convergence speed of them (I assume) are usually much slower for the majority of the problems, otherwise everyone should be using Newton method for optimizing their neural networks?

The authors included the wall clock calculation in the last page of the appendix, which is apperciated. In "time-critical applications" like "deciding whether to store or discard event recordings at the LHC", I highly doubt the physical gradient can be calculated using similar time as Adam, which is done using Auto-grad.

Would like to hear more about what other reviewers think as well. Maybe I am completely off and did not understand major part of it.

---

### Author Response · Authors · 2021-11-19
**Response to mue1's comment**

We thank mue1 for the quick reply to our comments. There are several misunderstandings that we’d like to clarify below.

> I think if the optimizer is well tuned, Adam should converge to as good as a result as the Newton method although using more iterations, since the loss surface is the same?

This is a central point of our paper: While this is correct for convex problems in theory, it turns out that there are huge differences in practice. The gradient descent with Adam can take **exponentially** longer than Newton’s method, making it infeasible for many applications (we thoroughly tested all optimizers, and observed this across a wide range of tests). The same holds for neural network training using Adam vs PGs. Identical networks trained with PGs can yield huge improvements in inference performance.

>  Second order information is definitely useful [...] [but] much slower for the majority of the problems, otherwise everyone should be using Newton method for optimizing their neural networks?

Second order information is expensive to obtain in network training tasks because the Hessian scales with the network weight count squared. This makes it impractical for learning but not for settings with fewer degrees of freedom (also see section 2.1 of our paper).

> In "time-critical applications" [...] I highly doubt the physical gradient can be calculated using similar time as Adam, which is done using Auto-grad.

When talking about time-critical applications, we are always referring to inference time, i.e. finding solutions x*. Here, networks trained with PGs yield identical execution times, but with the accuracy improvements mentioned in our paper. Network training, which is when the PGs are computed, happens offline. There, PGs are usually also approximately as fast as backpropagation.

> a lot of deep inverse models tries to learn directly the mapping from y to x space, see this below paper where it benchmarked a wide range of deep inverse solutions, this sounds like the "Tandem" model with a actual physical model replacing the forward proxy.

With tandem models, a neural network is first trained to approximate the forward process from data. However, in our setting, we assume the existence of a differentiable simulator for the forward process, so this additional abstraction step is not required.

---

> ### Comment · Reviewer_FRkr · 2021-11-20
> **Further discussion on mue1's comment**
>
> In response to mue1, I agree with all of the authors' points above. I think the paper is well-motivated and, after their revisions/clarifications, I now better understand the methodology. My only major remaining comment is with respect to the clarity of the overall manuscript.

---

> ### Comment · Reviewer_mue1 · 2021-11-20
> **Thanks for the quick response!**
>
> Thank the authors for their quick response, they are really helpful for my understanding.
>
> I still have question regarding the comparison with the deep learning inverse models.
>
> >> There are a lot of deep inverse models that are based on neural network. [...] See [...] Ren et al. in NeurIPS 2020
>
> >The methods you are referring to do not have access to the forward function but must learn from data alone. While the task discussed there is also very interesting, it poses completely different challenges than our setting.
>
> Since they only require samples of the labels, which can be trivially obtained if assuming you have access to the forward function, isn't that all those method can always be applied when this physical gradient method can be applied (and not the opposite?), thus making it necessary for this method to compare with them (since your setting can be trivially converted to their problem setting?)
>
> Feel free to leave quick, informal response to help with my ignorance : ) Happy to have more chat and learn more about this

---

> > ### Comment · Reviewer_FRkr · 2021-11-20
> > **Brief response**
> >
> > Quickly, if the forward function is expensive to run, then obtaining a supervised dataset can be extremely expensive! So in many domains, it is necessary to avoid this step. (Not all domains have a large number of "real-world examples" that can be obtained, which is what Ren et al. deal with.)

---

> > > ### Comment · Reviewer_mue1 · 2021-11-20
> > > **thank you reviewer FRkr for quick response！**
> > >
> > > I totally agree with the forward being expensive to run would cause significant trouble for deep inverse models. However, that would be also troublesome for method in this paper?
> > >
> > > For the result plots shown in this manuscripts, the network training all used equal or more than few thousands of simulation calls, which can not be parallelized due to the sequential nature of physical gradient algorithm while data-driven deep inverse models the simulation process can be highly parallelized due to i.i.d. assumption on training data.

---

> > > > ### Comment · Reviewer_FRkr · 2021-12-01
> > > > **clarifying my understanding**
> > > >
> > > > Sorry now for my delayed response! Maybe I'm misunderstanding, but I don't believe training requires forward simulation here -- rather, just a cheap approximation to the inverse solver. (Notably Equation 5, the equation the authors optimize, does not involve $\mathcal{P}$, but instead $\mathcal{P}^{-1}_n$.)

---

### Decision · Program_Chairs · 2022-01-20

**Decision:**

Reject

**Comment:**

The paper introduces a method to solve inverse problems: given y, find x such that P(x)=y, for a given physical simulator P. A standard approach is to learn a neural net such that the inverse x=NN(y;\theta). The authors state that this is problematic because it is difficult to take "higher order" gradient information into consideration when using this standard approach. The method assumes that there is an approximate inverse solver inv(P) and discusses an alternative "Physical Gradient" objective that can incorporate knowledge of an approximate inv(P) and a neural network. The experiments are good though comparing performance on an iteration basis is not always fair since an iteration of the PG method can be much more expensive than standard approaches.

The biggest issue that reviewers had was the clarity of the presentation. The authors have made a reasonable attempt to correct this, but I'm inclined to agree with the general reviewer sentiment that the presentation is still not at the required level. I agree that there are many things that are not clear, including the confusing discussion in section 2.1 about how the method takes higher order information into consideration. It only becomes partially transparent later in the experiments what is meant by higher order information.

Overall, I feel this is the basis of a potentially valuable contribution but that the current presentation is quite confusing. As mentioned by others, I would also suggest to find a different name since Physical Gradient is also rather misleading.

The following points were not part of the review process and I do not base the final decision on them, but the authors may want to consider the following:

I believe there is also an error in the basic approach, or at least an approximation is made which is not explained. The error is that the approximate inv(P) depends also on the parameter \theta (since this is used to initialise inv(P)). This dependency is not taken into consideration in the paper. For example, in theorem 1 in appendix A, the calculation of the gradient dM/d\theta is incorrect since the authors assume that inv(P) is independent of \theta, which it is not (since the preconditioner value depends on \theta). If we do take this into account, we would need to know the derivative of inv(P|x) with respect to the preconditioner x. This dependency would alter the gradient, potentially considerably. The gradient in figure 2 for the PG is also incorrect. One may of course simply say that the paper discounts this correction term in order to retain tractability; however, this would need to be stated as an approximation.